Manuscript prepared for Earth Surf. Dynam.
with version 2014/09/16 7.15 Copernicus papers of the LATEX class copernicus.cls.
Date: 26 May 2017

# Geomorphometric delineation of floodplains and terraces from objectively defined topographic thresholds

Fiona J. Clubb[1], Simon M. Mudd[1], David T. Milodowski[2], Declan A. Valters[3], Louise J. Slater[4], Martin D. Hurst[5], and Ajay B. Limaye[6]

[1]School of GeoSciences, University of Edinburgh, Drummond Street, Edinburgh, United Kingdom, EH8 9XP
[2]School of GeoSciences, University of Edinburgh, Crew Building, King's Buildings, Edinburgh, United Kingdom, EH9 3JN
[3]School of Earth, Atmospheric, and Environmental Science, University of Manchester, Oxford Road, Manchester, United Kingdom, M13 9PL
[4]Department of Geography, Loughborough University, Loughborough, United Kingdom, LE11 3TU
[5]School of Geographical and Earth Sciences, East Quadrangle, University of Glasgow, Glasgow, United Kingdom, G12 8QQ
[6]Department of Earth Sciences and St. Anthony Falls Laboratory, University of Minnesota, Minneapolis, Minnesota, USA

*Correspondence to:* Fiona J. Clubb (f.clubb@ed.ac.uk)

**Abstract.** Floodplain and terrace features can provide information about current and past fluvial processes, including channel response to varying discharge and sediment flux; sediment storage; and the climatic or tectonic history of a catchment. Previous methods of identifying floodplain and terraces from digital elevation models (DEMs) tend to be semi-automated, requiring the input of independent datasets or manual editing by the user. In this study we present a new method of identifying floodplain and terrace features based on two thresholds: local gradient, and elevation compared to the nearest channel. These thresholds are calculated statistically from the DEM using quantile-quantile plots and do not need to be set manually for each landscape in question. We test our method against field-mapped floodplain initiation points, published flood hazard maps, and digitised terrace surfaces from seven field sites from the US and one field site from the UK. For each site, we use high-resolution DEMs derived from light detection and ranging (LiDAR) where available, as well as coarser resolution national datasets to test the sensitivity of our method to grid resolution. We find that our method is successful in extracting floodplain and terrace features compared to the field-mapped data from the range of landscapes and grid resolutions tested. The method is most accurate in areas where there is a contrast in slope and elevation between the feature of interest and the surrounding landscape, such as confined valley settings. Our method provides a new tool for rapidly and objectively identifying floodplain and terrace features on a landscape scale, with applications

including flood risk mapping, reconstruction of landscape evolution, and quantification of sediment storage and routing.

## 1 Introduction

Identifying the location of floodplains and fluvial terrace features can provide important insights into geomorphic and hydrological processes. Understanding the controls on floodplain inundation carries increasing societal importance, as the frequency of flood events is predicted to increase with the rise in global temperatures and varying patterns of precipitation caused by climate change (Schreider et al., 2000; Booij, 2005; Hartmann et al., 2013). Although there are still large uncertainties regarding the impacts of climate change on flood frequency (Booij, 2005), identifying floodplains is crucial for forecasting and planning purposes. On longer timescales, the morphology and structure of fluvial terraces can provide important information on channel response to climatic, tectonic, and base-level variations (Bull, 1991; Merritts et al., 1994; Pazzaglia et al., 1998); the relative importance of lateral and vertical channel incision (Finnegan and Dietrich, 2011); and sediment storage and dynamics (Pazzaglia, 2013; Gran et al., 2013).

Attempts to identify floodplains can be classified into two broad families of methods: (i) flood risk mapping and hydrological modelling, and (ii) geometric terrain classification. Traditionally, identification of floodplains has relied upon the creation of flood hazard maps, produced through detailed hydraulic modelling studies (e.g. Noman et al., 2001; Grimaldi et al., 2013). These studies tend to incorporate historical flood event information, hydrological analyses, and hydraulic flow propagation models (Degiorgis et al., 2012). These mature techniques can lead to accurate flood inundation predictions down to the level of a single building (e.g. Horritt and Bates, 2002; Cobby et al., 2003; Guzzetti et al., 2005; Hunter et al., 2007; Kim et al., 2012). However, these models can be computationally expensive and time-consuming to run, even in one dimension, requiring the calibration of large numbers of parameters, all with their own uncertainties (e.g. Beven, 1993; Horritt and Bates, 2002; Liu and Gupta, 2007). This means that hydraulic simulations are usually performed at cross sections across the channel and interpolated to cover the rest of the stream network (Noman et al., 2001; Dodov and Foufoula-Georgiou, 2006). For example, floodplain mapping tools have been developed that incorporate either field-based or modelled stage-duration information at multiple cross sections along the channel, and interpolate a three-dimensional water surface between these sections (e.g Belmont, 2011; Yang et al., 2006).

The introduction of high-resolution digital elevation models (DEMs) has provided the opportunity to map floodplain features much more rapidly and over larger spatial scales than previously possible (Noman et al., 2001). This had led to the development of many different methods that rely on extracting a variety of topographic indices from DEMs, such as local slope, contributing area, and curvature (Manfreda et al., 2014). One common metric used to predict floodplains is the to-

pographic index ($\phi = ln(A/(tan\beta))$), where $A$ is the contributing area to each cell (m$^2$) and $\beta$ is the local slope in degrees (e.g. Kirkby, 1975; Beven and Kirkby, 1979; Beven et al., 1995; Quinn et al., 1995; Beven, 1997). The contributing area term reflects the tendency of water to accumulate at certain regions of the basin, whereas the slope term represents the tendency for gravity to transport water downhill. Therefore, high values of the topographic index represent areas which are likely to saturate first, as they have a large contributing area compared to local slope (Beven, 1997). Manfreda et al. (2011) suggested a modified version of the topographic index, changing the weighting on the area term by raising it to an exponent $n$. This modification allows the relative importance of slope or contributing area to be changed by varying the $n$ parameter. They proposed that floodplains can be identified as cells with a modified topographic index ($\phi_m$) greater than a threshold value, $\tau$. However, this method requires calibration of the parameters $\tau$ and $n$ through comparing the output floodplain map with a pre-existing hazard map, and noting the occurrence of true and false positives and negatives (Manfreda et al., 2011).

Another geometric method that has been developed to identify floodplains uses a series of linear binary classifiers for a number of topographic metrics (Degiorgis et al., 2012). Five different parameters are sampled from the DEM (slope, contributing area, elevation from nearest channel, distance from nearest channel, and curvature), and each cell is classified as either 1 (floodplain) or 0 (non-floodplain) depending on whether these parameters are above or below threshold values. Each of these five metrics can be considered in isolation or in pairs. The thresholds are calibrated using flood hazard maps, where the number of true and false positives and negatives are noted, similar to the approach of Manfreda et al. (2011). For each parameter and threshold value the Receiver Operating Characteristics (ROC) curve (e.g. Fawcett, 2006) is calculated, which is defined by the number of true and false positives. The maximum area under the curve is determined to allow the threshold value for each parameter to be calibrated, as well as comparisons between each parameter to be found. The pair of best-performing features was identified as the distance (D) and elevation (H) from the nearest channel (m). This method is also semi-automated, as it requires the existence of flood hazard maps for at least some part of the catchment in order to select the correct binary classifiers for floodplain identification.

Dodov and Foufoula-Georgiou (2006) present an algorithm for identifying floodplains over large scales based on information on bankfull channel depths. They suggest that the morphology of the floodplain is defined by the lateral channel migration rate through time, and is controlled by the transport of water and sediment by the channel. Therefore, they assume that the geometry of the floodplain is related to that of the channel, and demonstrate a relationship between bankfull channel depths and floodplain inundation depths which is linear over a range of scales (Dodov and Foufoula-Georgiou, 2006). Floodplain delineation is carried out by locally filling the DEM up to the depth of inundation, which is determined based on bankfull channel depths, calibrated using data from United States Geological Survey (USGS) gauging stations across Oklahoma and Kansas, along with

field measurements. The depth of inundation at points along the channel network is then used to find the lateral extent of the floodplain by using the planform curvature of the channel. This method also requires significant user input, as the channel bankfull depths are required in order to estimate the inundation depth.

The extraction of fluvial terraces (the remnants of previous floodplains) represents a closely related
problem to the delineation of presently active floodplain surfaces. Previous studies have also used a geometric approach to identify terrace features from DEMs. For example, Demoulin et al. (2007) identified terrace surfaces based on local slope and height of each pixel compared to the channel. They used these attributes in order to reconstruct palaeo-channel profiles from terrace surfaces, but their methodology was not designed to produce a map of terrace extents on a wider landscape scale.
Therefore, following on from their approach, Stout and Belmont (2014) presented the TerEx toolbox, a semi-automated tool to identify potential terrace surfaces based on thresholds of local relief, minimum area, and maximum distance from the channel. After potential terrace surfaces are identified, their area and height above the local channel are measured. The tool then allows the user to edit the terrace surfaces based on comparison with field data. Hopkins and Snyder (2016) evaluated
the TerEx toolbox, along with two other semi-automated methods for identifying terrace surfaces (Wood, 1996; Walter et al., 2007) at the Sheepscot River, Maine. They found that all of the methods over-predicted terrace areas compared to the field-mapped terraces, and the accuracy of the methods decreased in lower relief landscapes. These semi-automated methods allow the user to manually clip over-predicted terrace surfaces based on field data and DEM observations, and remove selected sur-
faces that do not represent terraces, such as roads, alluvial fans, or water bodies Stout and Belmont (2014).

The geomorphic methods of mapping both terraces and floodplains outlined above are all semi-automated, requiring independent datasets and significant user input. For example, the method proposed by Manfreda et al. (2011) requires the parameters to be optimised using flood inundation
maps from hydraulic simulations. The linear binary classifiers outlined by Degiorgis et al. (2012) and tested by Manfreda et al. (2014) use flood hazard maps to select the correct threshold for floodplain prediction from the geomorphic indices. The TerEx toolbox, developed by Stout and Belmont (2014), requires significant user input in order to manually edit the predicted terrace surfaces. No existing approach to mapping either floodplains or terraces from topographic data includes objec-
tive criteria for setting the thresholds that identify floodplains and terraces. As a result, the different thresholds that a user might select can result in varying floodplain and terrace maps for the same input DEM, complicating efforts to consistently map geomorphic features between different landscapes.

Here we introduce a new method of identifying floodplain and terrace surfaces from topographic
data. This method uses two geometric thresholds that can be readily extracted from DEMs: the gradient of each pixel, and the elevation of each pixel relative to the nearest channel. Importantly, this

method does not require calibration using any independent datasets, as the thresholds are statistically calculated from the DEM using quantile-quantile plots. We test our method against field-mapped floodplain initiation points, published flood hazard maps, and digitised terrace surfaces from seven field sites throughout the US and one site in the UK (Figure 1). For each site, where available, we use high-resolution LiDAR-derived DEMs, as well as the corresponding national elevation datasets (10 m resolution for the US and 5 m for the UK) in order to test the sensitivity of our method to grid resolution.

## 2 Methodology

Floodplain and terrace surfaces can be defined as low relief, quasi-planar areas capped by alluvium and found proximal to the modern river channel. Therefore, field mapping campaigns typically identify these surfaces as spatially continuous areas with low gradients that occur next to the channel. We present a new geometric method which replicates this field approach as closely as possible by using two metrics which can be readily extracted from the DEM: elevation compared to the nearest channel, and local gradient. Our method is efficient to run and is based on the statistical selection of topographic thresholds, requiring no input of independent datasets or field mapping. We outline below the DEM pre-processing steps followed by the methodology for identifying floodplain and terrace features.

### 2.1 DEM pre-processing

The first step of the algorithm is to smooth the DEM in order to remove micro-topographic noise. Gaussian filters are often used to smooth DEMs, where the smoothing can be described by linear diffusion. A Gaussian filter results in the DEM being smoothed uniformly at all locations and in all directions (e.g. Lashermes et al., 2007). However, one consequence of the Gaussian filtering is the loss of information where there are sharp boundaries between features due to the uniform smoothing. Therefore, we filter the input DEM using a non-linear filter proposed by Perona and Malik (1990), and applied to channel extraction from high-resolution topography by Passalacqua et al. (2010a). The Perona-Malik filter is an adaptive filter in which the degree of smoothing decreases as topographic gradient increases (Perona and Malik, 1990; Passalacqua et al., 2010a). This non-linear diffusion equation can be described as:

$$\partial_t h(x,y,t) = \triangledown.[p(|\triangledown h|)\triangledown h] \tag{1}$$

where $h$ is the elevation at location $(x, y)$ and time $t$, $\bigtriangledown$ is the gradient operator, and $p(|\bigtriangledown h|)$ is an edge-stopping function that specifies where to stop diffusion across feature boundaries, where:

$$p(|\bigtriangledown h|) = \frac{1}{1 + (|\bigtriangledown h| / \lambda)^2} \qquad (2)$$

where $\lambda$ is a constant. Importantly for the identification of low-gradient surfaces, the Perona-Malik filtering enhances the transitions between features, such as the low-gradient valley floor and the surrounding hillslopes, while preferentially smoothing low gradient reaches of the DEM. Following the methodology of Passalacqua et al. (2010a), we set the number of iterations ($t$) to 50 and the calculation of $\lambda$ as the 90% quantile. We keep these parameters constant across each site tested in the study. A full explanation of these parameters and derivation of the Perona-Malik filter is described by Passalacqua et al. (2010a).

After the DEM is smoothed, we then extract the channel network. Many studies have proposed different methods for identifying channel networks from high-resolution topography (e.g. Lashermes et al., 2007; Tarolli and Dalla Fontana, 2009; Passalacqua et al., 2010b, 2012; Pelletier, 2013; Clubb et al., 2014). Grieve et al. (2016c) tested the validity of channel extraction algorithms at coarsening DEM resolution, and found that a geometric method of channel extraction was consistent up to DEM resolutions of 30 m. This method, described by Grieve et al. (2016b), uses an Optimal Wiener filter to remove micro-topographic noise from the DEM (Wiener, 1949; Pelletier, 2013). The Optimal Wiener filter is only used to extract the channel network: we use the Perona-Malik filtering to extract the floodplains and terraces. Channelised portions of the landscape are selected using a tangential curvature threshold (Pelletier, 2013), which is defined using quantile-quantile plots as described by Lashermes et al. (2007) and Passalacqua et al. (2010a). These channelised portions of the landscape are combined into a channel network using a connected components algorithm outlined by He et al. (2008), and thinned using the algorithm of Zhang and Suen (1984). We chose this algorithm for channel extraction to allow consistency when running our method on DEMs of varying grid resolutions.

## 2.2 Floodplain and terrace identification

After smoothing the DEM, the user can choose to run the terrace and floodplain mapping algorithm across the whole DEM, or to extract the floodplains and terraces relative to a specific channel of interest. If the algorithm is run on the whole DEM, the local gradient, $S$, and relief relative to the nearest channel, $R_c$, are calculated for each pixel. These two parameters were chosen on the basis that floodplains and terraces tend to form low-gradient regions that are close to the elevation of the modern channel. Local gradient has been used in previous geometric methods of floodplain and terrace identification, both in the calculation of the topographic index (Kirkby, 1975; Manfreda et al., 2011), and in combination with other topographic metrics (e.g. Degiorgis et al., 2012; Stout

and Belmont, 2014; Limaye and Lamb, 2016). Local gradient was calculated by fitting a polynomial surface to the DEM with a circular window (e.g. Lashermes et al., 2007; Roering et al., 2010; Hurst et al., 2012; Grieve et al., 2016a). The radius of the window is calculated by identifying breaks in the standard deviation and interquartile range of curvature with increasing window size, following Grieve et al. (2016a). This allows the window size to be calculated for each DEM to ensure that

the slope values are representative at the hillslope scale, rather than being influenced by smaller-scale variations from vegetation (e.g. Roering et al., 2010; Hurst et al., 2012). $R_c$ has also been used in previous geometric methods (e.g. Degiorgis et al., 2012; Manfreda et al., 2014; Limaye and Lamb, 2016), and is calculated as the difference in elevation between the starting pixel and the nearest channel pixel, identified using a steepest descent flow routing algorithm (O'Callaghan and

Mark, 1984; Braun and Willett, 2013). A threshold Strahler stream order is set by the user such that the nearest channel must have a stream order greater than the threshold. This is necessary so that each pixel is mapped to the main channel along which floodplains or terraces have formed, rather than narrow tributary valleys. We found that a threshold of third order channels was appropriate for each of our field sites, based on a visual inspection of the DEM. One of the outputs of our software

package is a raster of the channel network labelled by the Strahler stream order. The user can identify an appropriate threshold stream order based on visual inspection of floodplain and terrace surfaces compared to this network.

As well as running the algorithm on the whole landscape, the user can also choose to extract floodplains or terraces relative to a specific channel of interest. The user must provide the latitude

and longitude of two points defining the upstream and downstream end of the channel. The algorithm then defines a channel network between these points using a steepest descent flow routing algorithm (O'Callaghan and Mark, 1984; Braun and Willett, 2013). After the identification of the channel, a swath profile is created along it following the method outlined in Hergarten et al. (2014) and applied by Dingle et al. (2016). The user must specify the width of the swath, which can be estimated by

a visual inspection of the DEM, to provide a sufficiently wide swath compared to the valleys in the landscape. The same two parameters ($S$ and $R_c$) are used for feature classification for each pixel in the swath profile, except that $R_c$ is calculated compared to the nearest point on the reference channel.

After the calculation of slope and $R_c$, we identify thresholds for each metric in order to provide a binary classification of each pixel as either floodplain/terrace (1) or hillslope (0). A key feature

of our new method is that the thresholds for $R_c$ and local gradient do not need to be set by the user based on independent validation, but are calculated statistically from the DEM. Many methods of channel extraction employ statistical selection of topographic thresholds (e.g. Lashermes et al., 2007; Thommeret et al., 2010; Passalacqua et al., 2010a; Pelletier, 2013; Clubb et al., 2014), but this has yet to be developed for the identification of floodplains or terraces. We identify thresholds for $R_c$

and $S$ using quantile-quantile plots, which have previously been used in the detection of hillslope-valley transitions (e.g. Lashermes et al., 2007; Passalacqua et al., 2010a). Quantile-quantile plots

are used to determine if a probability density function of real data can be described by a Gaussian distribution. The transition between process domains can be determined by the value at which the probability density function of the real data deviates from the Gaussian function (Lashermes et al., 2007). The real data are plotted against the corresponding standard normal variate, which indicates how many standard deviations an element is from the mean. For example, if a value has a standard normal variate (or z-score) of 1, then it is one standard deviation above the mean, which has a z-score of 0. A Gaussian distribution plots as a straight line on a quantile-quantile plot, and is modelled for each DEM based on a lower and upper percentile of the real data. The percentiles chosen to represent the reference Gaussian distribution can be set by the user based on the landscape in question, but are generally set as the 25$^{th}$ and 75$^{th}$ percentile (Passalacqua et al., 2010a). For each value of the real data, we calculate the difference between the real data and the Gaussian distribution as a fraction of the range of the real data (Figure 2). The threshold values for $R_c$ and slope are then identified as the lowest value at which there is less than 1% difference between the two distributions. Figure 3 shows an example of the channel relief and slope maps for the Russian River field site, with the calculated thresholds for each field site presented in Table 1. If the user wishes to extract only the terraces, then a threshold height above the modern river channel must be set: any pixels below this height will be identified as floodplain, and any pixels above this height will be identified as terraces. This threshold height can also be determined based on a visual inspection of the DEM. Our method allows the analysis of spatial extent of floodplain and terrace features (if run across the whole DEM) as well as the distribution along a specific channel of interest (if run with the swath mode). For example, in swath mode, the elevation and slope of the terraces can be mapped as a function of distance upstream along the channel network. This provides numerous potential applications of the method for understanding controls on terrace formation and morphology.

## 2.3 Comparison with published data

In order to test the results of our method we compare the predicted floodplain and terrace locations to field-mapped floodplain initiation points, published flood hazard maps, and digitised terrace surfaces. In order to quantify the performance of our methods compared to these datasets, we assess the rates of true positives ($TP$), false positives ($FP$), true negatives ($TN$), and false negatives ($FN$) (e.g. Heipke et al., 1997; Molloy and Stepinski, 2007; Tarolli et al., 2010; Orlandini et al., 2011; Manfreda et al., 2014; Clubb et al., 2014). Each pixel is assigned to one of the four categories:

1. True positive $TP$: The pixel is identified as floodplain/terrace by both the geomorphic method and the independent dataset.

2. False positive $FP$: The pixel is identified as floodplain/terrace by the geomorphic method, but not by the independent dataset.

3. True negative $TN$: The pixel is not identified as floodplain/terrace by either dataset.

4. False negative $FN$: The pixel is identified as floodplain/terrace by the independent dataset but not by the geomorphic method.

We report the reliability ($r$), sensitivity ($s$), and overall quality ($Q$) for each field site:

$$r = \frac{\sum TP}{\sum TP + \sum FP} \tag{3a}$$

$$s = \frac{\sum TP}{\sum TP + \sum FN} \tag{3b}$$

$$Q = \frac{\sum TP}{\sum TP + \sum FP + \sum FN} \tag{3c}$$

The reliability, $r$, is a measure of the ability of the method to not generate false positives. The $r$ value can vary between 0 and 1: if the $r$ value is low, then the method is predicting a large amount of pixels as floodplain or terrace which are not identified by the independent dataset, whereas as high $r$ value indicates that the majority of pixels mapped as floodplain or terrace are also identified by the independent map. The sensitivity, $s$, is a measure of the ability of the method to not generate false negatives: a low $s$ value indicates that the method is not identifying many of the floodplain or terrace pixels selected by the published maps. The overall quality, $Q$, combines both the number of false positives and false negatives to give an overall 'goodness' of the feature classification. It also varies between 0 and 1, where 0 represents no correlation between the predicted and observed features, and 1 represents a perfect match (Heipke et al., 1997).

## 3 Study areas

We ran our new method on a total of eight field sites, located in Figure 1. Four of these field sites (the Russian River, CA; Mid Bailey Run, OH; Coweeta NC; and the River Swale, UK) were selected to test the ability of the algorithm to identify floodplains, using published flood maps for the regions. The remaining four sites were selected to validate the algorithm against digitised terrace maps (South Fork Eel River, CA; Le Sueur River, MN; Clearwater River, WA, and Mattole River, CA). Table 2 summarises the mean annual precipitation and mean annual temperature of each site, based on data from the PRISM Climate Group (http://prism.oregonstate.edu) for the US sites and the Met Office (http://www.metoffice.gov.uk/public/weather/climate/) for the UK site. It also summarises the underlying lithology, the source of the data used for validation, and the grid resolution. The algorithm was run based on topographic data derived from 1 m LiDAR data for the sites where these were available (the Russian River, CA; Mid Bailey Run, OH; Coweeta, NC; the South Fork Eel River, CA; and the Le Sueur River, MN). For the remaining field sites the topographic data were generated from the United States Geological Survey National Elevation Dataset 1/3 arc sec DEM, sampled at 10 m resolution for the US sites, and from the Ordnance Survey Terrain 5 dataset for the

UK site, sampled at 5 m resolution. All DEMs were converted to the Universal Transverse Mercator (UTM) coordinate system using the WGS84 datum.

## 4 Results

### 4.1 Comparison with mapped floodplains

We compare the floodplain extent predicted by the our method to field mapped floodplain initiation points (FIPs) from two of the four study areas: Mid Bailey Run, OH, and Coweeta, NC. A FIP was defined as the upstream limit of low gradient surfaces at the same elevation as the channel banks. As the valley opens out from its more confined upper reaches, these surfaces transition from discontinuous depositional pockets to more continuous floodplain surfaces (Jain et al., 2008). In this study we consider the FIP to start at the onset of alluviation outside the channel banks: therefore, we mapped the start of the discontinuous floodplain pockets at the FIPs in each channel. The onset of alluviation often occurred at multiple locations along the same channel: in these cases we took the location of each FIP downstream along the channel.

A total of 19 FIPs were mapped in Mid Bailey Run, OH, during May–June 2011, and eight FIPs were mapped in the Coweeta catchment, NC, in May 2014. FIPs in the Mid Bailey Run catchment were mapped using a Trimble GeoXM GeoExplorer 2008 series GPS with a mean horizontal accuracy of 6 m. Point locations in the Coweeta catchment were mapped using a Trimble GeoXR GeoExplorer 6000 series GPS with a mean horizontal accuracy of 1.01 m and a mean precision of 1.3 m. Figure 4 shows the relationship between the field mapped initiation points and predicted floodplain extent. In order to compare these field mapped FIPs to our predicted floodplain extents, we measured the flow distance between the field mapped point and the furthest upstream point of the nearest predicted floodplain patch. The distances for each FIP are reported in Table 3, where negative values indicate that the predicted floodplain initiation was upstream of the mapped, and vice versa for positive values. We also report the $r$, $s$, and $Q$ values for the predicted floodplain initiation points. Following the methodology of Orlandini et al. (2011), we classify a point as a TP if the predicted FIP is within a 30 m radius of the mapped FIP. The comparison with the mapped FIPs resulted in $r = 0.83$, $s = 0.67$, and $Q = 0.59$ for Mid Bailey Run, and $r = 0.78$, $s = 1$, and $Q = 0.78$ for Coweeta.

Along with these field mapped floodplain initiation points, we also compare our predicted floodplain extent to published flood risk maps for three out of the four study areas. For the sites in the US, flood risk maps were obtained from the Federal Emergency Management Agency (FEMA)'s National Flood Hazard Layer (https://msc.fema.gov/portal/). The National Flood Hazard Layer is a compilation of GIS data consisting of a US-wide Flood Insurance Rate map. It contains information on the flood zone, base flood elevation, and floodway status for a location. Floodplain extents are calculated using a hydraulic model, such as HEC-RAS (Hydrologic Engineering Center-River

Analysis System), incorporating discharge data, cross sectional survey data, and stream characteristics. These studies can be expensive, with a detailed survey on a mile-long reach typically costing between \$10,000 and \$25,000 (Committee on FEMA Flood Maps, 2009). The original data were in the geographic projection NAD1983, and were converted to the projected UTM WGS84 coordinate system (Ohio and NC Zone 17N, Russian River Zone 10N). We separate the flood zones into two categories: areas within the 100 year flood (blue), with a 1% annual chance of flooding, and areas with a greater than 100 year flood risk (less than 1% annual risk of flooding). In order to compare these maps to our method, we gridded the FEMA flood risk maps with a resolution of 1 m. The Coweeta field site in North Carolina did not have a complete flood risk map for the catchment and therefore could not be included in this analysis.

For the River Swale field site in the UK, flood risk maps were obtained from the Environment Agency's (EA) Risk of Flooding from Rivers and Sea dataset, which divides the landscape into 50 by 50 m cells (https://data.gov.uk/dataset/risk-of-flooding-from-rivers-and-sea1). Each cell is categorized into one of four flood risk likelihood categories: high (3.3% annual chance of flooding); medium (between 3.3% and 1%); low (between 1% and 0.1%); or very low (<0.1%). The dataset is created by hydraulic modelling, including information about the state of flood defenses and local stage heights as inputs to the model. The data were re-projected from the British National Grid coordinate system to the UTM WGS84 datum, Zone 30N. In order to keep the comparison consistent with the sites from the US, each pixel was classified into the same two categories as for the FEMA maps, with areas of flood risk identified as having greater than 1% annual chance of flooding. The dataset is provided as vector data: to compare with the floodplain identified by the our method, we gridded the vector dataset at 5 m resolution (the same as the input DEM). Figure 5 shows examples of the FEMA and EA flood maps for each study area.

The $r$, $s$, and $Q$ values for each site are reported in Table 4, with a visual comparison between the method and the published flood maps shown in Figure 6. We also report the quality values for floodplains extracted from the United States Geological Survey's 1/3 arc second National Elevation Dataset (NED), gridded at 10 m, in order to test the sensitivity of our method to grid resolution. The USGS NED is a seamless dataset created for the conterminous US, using a variety of elevation products which is updated on a two-month cycle. The method was most similar to the flood risk maps for the Russian River, CA with the highest overall quality value ($Q = 0.67$ for the 1 m DEM and $0.68$ for the 10 m DEM). The method has a higher sensitivity than reliability for both DEM datasets, with $s = 0.97$ and $r = 0.74$ for the 1 m DEM; compared to $s = 0.96$ and $r = 0.70$ for the 10 m DEM. For both the Mid Bailey Run and Russian River field sites, the sensitivity is higher than the reliability for all of the DEM resolutions tested (Table 4). However for the River Swale site, the reliability is higher than the sensitivity ($r = 0.84, s = 0.65$).

## 4.2 Comparison with mapped terraces

We also compare the features extracted by our method to field-mapped terraces from four field sites throughout the US: the South Fork Eel River, CA (Seidl and Dietrich, 1992); the Le Sueur River, MN (Gran et al., 2009); the Mattole River, CA (Dibblee and Minch, 2008); and the Clearwater River, WA (Wegmann and Pazzaglia, 2002). Two of these sites had 1 m LiDAR-derived DEMs (the South Fork Eel and Le Sueur Rivers). For the remaining two sites, 10 m DEMs were derived from the USGS 1/3 arc second NED, following Limaye and Lamb (2016). Terraces in the South Fork Eel River and the Le Sueur River were digitised from field mapping carried out in previous studies (Seidl and Dietrich, 1992; Gran et al., 2009), constrained by the hillshaded DEMs. Terraces from the Mattole River and the Clearwater River were digitised by Limaye and Lamb (2016) from geological maps, with the terraces mapped by Dibblee and Minch (2008) for the Mattole River, and Wegmann and Pazzaglia (2002) for the Clearwater River. We ran our method in the swath setting for each of these sites, so that the terraces were mapped compared to the main stem channel of interest in each site. The thresholds for terrace identification ($R_c$ and $S$) were set statistically for each site using the quantile-quantile plots. In order to quantify the difference between our method and the digitised terraces, we calculated the $r$ and $s$ values following the same methodology as for the floodplain comparison (Table 4).

Figure 7 shows a visual comparison of the predicted and digitised terraces from the two sites with 1 m LiDAR-derived DEMs. In general there was good spatial correlation between the two terrace datasets for each field site, although in some cases the automated method did not identify all terraces at high elevations compared to the modern channel. The South Fork Eel River had the highest values of both $r$ (0.65) and $s$ (0.72). The comparison between the two terrace datasets for the field sites with 10 m DEMs is shown in Figure 8. These sites had lower $r$ and $s$ values than that of the South Fork Eel River, but were comparable to the values for the Le Sueur River (e.g. Table 4).

## 5 Discussion

### 5.1 Floodplains

The results outlined above compare our method of automatic feature extraction to various datasets of both floodplains and terraces. In order to test the ability of our method in identifying floodplains, we compared the delineated geomorphic floodplain to both field-mapped floodplain initiation points and hydrological modelling predictions. We found that our method predicts the location of the field-mapped FIPs to within tens of metres for both field sites (Mid Bailey Run, OH; and Coweeta, NC). The reliability and sensitivity values were highest for the Coweeta field sites, with a value of $r = 0.78$ and $s = 1$, which indicates that there were no false negatives in this field site. Table 3 shows that in many cases the error between the mapped and predicted FIPs is within the same order of

magnitude as the error on the field-mapped coordinates ($\approx 1$ m for Coweeta and $\approx 6$ m for Mid Bailey Run). In isolated cases in the Mid Bailey Run site, the error was higher between the mapped and predicted FIPs (around 90 m for two of the points), where the mapped FIP was located in

narrow headwater valleys (Figure 4). Furthermore, the predicted floodplain in the majority of cases was located downstream of the mapped FIPs in Mid Bailey Run (Table 3). This is not surprising, as our method is based on identifying areas of low gradient, which is calculated based on polynomial surface fitting with a specified window radius (Sect. 2.2). Small pockets of alluviation in narrow valleys may therefore be missed by the method if the width of the floodplain is less than that of the

window radius or the DEM resolution.

We also validated our method against published flood maps for three of our field sites (Mid Bailey Run, OH; Russian River, CA; and River Swale, UK). The quality analysis for this comparison (Table 4 and Figure 6) suggests that there is in general a good correlation between our method and the published flood maps, with high values for reliability ($r \geq 0.7$), sensitivity ($s \geq 0.65$), and overall

quality ($Q \geq 0.58$) for each field site. The results for both the Russian River and Mid Bailey Run showed higher sensitivity values than reliability, suggesting that the our method predicted more false positives than false negatives. In each field site, the published flood maps were classified to define the 1% annual chance of flooding, or the 100 year return period flood event. It may therefore be expected that our geomorphic-based method would delineate a larger floodplain than is flooded in

a 100 year return period event. The results for the River Swale, however, show a higher reliability than sensitivity, suggesting that more false negatives were predicted than false positives. This may be due to methodological differences in the production of this flood map by the Environment Agency (UK) compared to the US sites. Figure 6f shows the published flood map for the River Swale site which, in comparison to the FEMA flood maps (Figures 6b and 6d) extends into the headwaters of

the channel network. As these areas do not have low gradient surfaces next to the channel, they may not be selected by our method. This may account for the higher number of false negatives predicted at this site.

Published flood maps are useful in providing an independent estimate of likely floodplains in each field site. However, there are potential limitations to these maps which must be carefully considered,

and may result in some of the differences compared to geomorphic floodplain prediction techniques. Hydrodynamic models have a large number of parameters, which require careful calibration with field and hydraulic data, such as channel roughness and discharge data from gauging stations. Furthermore, due to the time-consuming and expensive nature of these studies, flood maps are often not produced for small catchment sizes, and may therefore be incomplete on a landscape-scale (e.g.

Figure 5). There may also be differences in the methodology used in producing these maps for each site, depending on the input topographic data and modelling software used. However, despite these discrepancies between the flood maps we find a good spatial correlation between these and the predictions from our method (Figure 6).

In order to test the sensitivity of our method to grid resolution, we also ran the floodplain extraction using 10 m DEMs derived from the USGS NED for two of the field sites (Russian River, CA, and Mid Bailey Run, OH), as well as testing it on the River Swale in the UK (5 m resolution DEM). We found there was little difference in the reliability and sensitivity results when compared to the 1 m DEMs (Table 4). This suggests that our method is relatively insensitive to grid resolution, allowing the identification of floodplain features on coarser-resolution DEMs. Furthermore, in the Mid Bailey Run field site, the method performed better on the 10 m data compared to the 1 m DEM. High-resolution topographic data may contain both small-wavelength topographic noise caused by tree throw and biotic activity (Roering et al., 2010; Marshall and Roering, 2014), as well as synthetic noise from point cloud processing (Liu, 2008; Meng et al., 2010). This noise may affect the calculation of topographic metrics (Grieve et al., 2016c), potentially leading to differences in the location of extracted floodplains or terraces compared to the lower resolution data.

## 5.2 Terraces

We also tested the ability of our method to identify fluvial terraces in four field sites (South Fork Eel River, CA; Le Sueur River, MN; Mattole River, CA; and Clearwater River, WA) by comparing to digitised terrace maps. Two of these field sites had 1 m LiDAR-derived DEMs (Figure 7) whereas two had 10 m DEMs from the USGS NED (Figure 8). The quality analysis for the 1 m DEMs showed the higher reliability and sensitivity values for the South Fork Eel River site ($r = 0.65$ and $s = 0.72$), with comparable values for the remaining three field sites. This may be due to the influence of topographic structure on terrace identification. The portion of the Eel River DEM analysed here has higher relief, with a maximum elevation of 290 m above the nearest channel, compared to the lower-relief landscape covered by the DEM for the Le Sueur River, with a maximum elevation of 40 m above the nearest channel. As our method relies on the distribution of relief relative to the channel in order to select the threshold for terrace identification, it will work best in areas where there is a greater contrast between the slope and relief of the terrace surfaces compared to the surrounding topography, such as steep mountainous areas. This is similar to other semi-automated terrace extraction methods (e.g. Stout and Belmont, 2014; Hopkins and Snyder, 2016). The Le Sueur River is currently incising through Pleistocene tills, forming a low-gradient surface or plateau (Fisher, 2003; Gran et al., 2009; Belmont et al., 2011a). High-altitude, low-gradient surfaces, such as relict plateaus, may result in error in the method due to the difficulty in distinguishing the distribution of terrace elevations from these low-relief surfaces. The Le Sueur River basin is also heavily influenced by human land use, which makes feature extraction challenging (Passalacqua et al., 2012). The results of the quality analysis for the eight field sites (Table 4) showed that the method performed better in the floodplain identification compared to the terrace identification. This may be due to the fact that, with the exception of the South Fork Eel River, the sites used for terrace extraction are lower relief than those used to test the floodplain extraction (e.g. Figures 6 - 8).

Another potential cause of error between the predicted and digitised terrace locations may be prob-
lems in distinguishing whether features represent the modern floodplain or terraces. In our method
a minimum height above the modern channel is set, where pixels above this height are classified
as terrace, and below this height are classified as floodplain. In some cases, particularly where the
terraces are at a similar elevation to that of the modern channel, our method may mistakenly identify

terraces as being part of the modern floodplain, or vice versa. An example of this may be the Clear-
water River site, where our method had lower indices of $r$ and $Q$ (Figures 8c and d and Table 4). In
this site, the digitised terraces are close in elevation to the modern channel, with a maximum terrace
height of 13 m. Furthermore, in some cases our method did not select all of the terraces identified by
the field mapping, particularly at the highest elevations compared to the modern channel (e.g. Figure

7c and d). This may be the case if the threshold for elevation compared to the channel selected by
the quantile-quantile plot is lower than that of the highest terrace elevations. This can be examined
for the landscape in question by a visual inspection of the quantile-quantile plots and the location of
the threshold compared to the distribution of channel relief (e.g. Figure 2). Our method fits a Gaus-
sian distribution to the quantile-quantile plots, and selects the thresholds as the deviation of the real

data from this distribution, as a simple general model of elevation distributions that can be applied
across multiple landscapes. However, in some landscapes, the distribution of elevations may not be
accurately represented by a Gaussian distribution. A future avenue for development of this method
may be to include multiple models for elevation distributions from which to select the thresholds of
elevation and gradient.

However, despite these limitations, the selection of the threshold from quantile-quantile plots
means that our method does not require the input of any independent datasets or field-mapping.
Semi-automated methods of terrace identification, where the terrace polygons are manually edited by
the user, are particularly useful in areas where independent datasets of terrace locations are available
for calibration, and may be more appropriate than our method on site-specific scales (e.g. Stout and

Belmont, 2014). However, the selection of thresholds based on a statistical approach means that our
method can be applied in areas where these data do not exist, on a broader landscape scale, or as a
rapid first-order predictor of terrace locations.

    In addition to the field sites with LiDAR-derived DEMs, we also tested our method against digi-
tised terraces from two sites with 10 m DEMs gridded from the USGS NED, to examine the perfor-

mance of the method at lower grid resolution. Figure 8 shows the results of the terrace identification
on the 10 m resolution data. The reliability and sensitivity of the method for these two sites (Table 4)
was lower than that of the South Fork Eel River, but comparable to that of the Le Sueur River. This
suggests that the method is able to successfully select terraces at lower grid resolutions. Although
there are some differences between the terraces predicted by the method and those digitised in the

field, the majority of the terrace features evident from a visual inspection of the hillshaded DEMs
are correctly identified by the algorithm (Figure 8). In some cases, some terrace-like features that

can be seen on the hillshaded DEMs are not identified in the digitised terrace maps (e.g. Figure 8b). This may be due to error in the mapping of terrace surfaces in the field, or discrepancies resulting from the digitisation process.

An objective, landscape-scale method of identifying floodplain and terrace features has numerous applications in the geomorphological and hydrological communities. For example, terrace surfaces have been used to examine the response of fluvial systems to tectonic and climatic perturbations (e.g. Merritts et al., 1994), and to investigate the relative importance of lateral and vertical channel incision (e.g. Finnegan and Dietrich, 2011). Analysis of terrace areas can be used to quantify sediment

budgets and estimate storage volumes over millenial timescales (e.g. Trimble, 1999; Brown et al., 2009; Belmont et al., 2011b; Blöthe and Korup, 2013). Our new method facilitates the rapid extraction of terrace surfaces either across the whole landscape or compared to a representative channel of interest. It allows the user to investigate how various metrics, such as elevation compared to the channel, slope, and curvature, vary both within and between individual terrace surfaces (e.g. Figure

7). These metrics could be used in order to examine how terrace heights vary with distance along channel profiles, for example, or to identify signatures of deformation corresponding to tectonic processes (Avouac and Peltzer, 1993; Lavé and Avouac, 2000; Pazzaglia and Brandon, 2001; Viveen et al., 2014).

### 5.3   Research needs: fully-automated feature extraction

A key goal for the Earth surface research community is to develop fully-automated methods of feature extraction from DEMs in order to avoid expensive and time consuming field-mapping, and to investigate the controls on geomorphic processes at a landscape scale. Our new method of floodplain and terrace delineation attempts to meet some of these research needs, by allowing the statistical determination of the thresholds for feature extraction. However, our method still requires the input

of some user-defined parameters. If the method is run across the whole landscape, the user must set a threshold stream order for the calculation of elevation compared to the nearest channel. This is necessary so that each pixel is mapped to the main channel along which floodplains or terraces have formed, rather than narrow tributary valleys. This threshold can be determined by the user based on a visual inspection of the DEM compared to the channel network. If the user runs the method based

on the swath mode, the width of the swath profile must be set. This can also be done based on a visual inspection of the DEM to provide a sufficiently wide swath compared to the valleys in the landscape. Furthermore, if the method is run in the swath mode, then a minimum terrace height must be set in order to delineate between floodplains and fluvial terraces.

    However, future development of new algorithms, such as extraction of valley widths, would allow

these parameters to be set based on the topographic data alone. Our method represents a first step towards this goal of fully-automated geomorphic feature identification, which can be improved upon with future research. The combination of different algorithms for terrain analysis, such as hillslope

flow routing, channel network extraction, floodplains, and fluvial terraces, would allow an objective landscape-scale investigation of the controls on geomorphic processes.

## 6 Conclusions

We have presented a novel method for the geomorphometric delineation of floodplain and fluvial terrace features from topographic data. Unlike previous methods, which tend to require calibration with additional datasets, our method selects floodplain and terrace features using thresholds of local gradient and elevation compared to the nearest channel, which are calculated statistically from the DEM. Furthermore, the floodplain or terrace surfaces do not need to be manually edited by the user at any point during the process. Our method can be run either across the whole landscape, or from a topographic swath profile where features can be compared to a specific channel of interest.

In order to test the performance of our method we have compared it to field-mapped floodplains and terraces from eight field sites with a range of topographies and grid resolutions. We find that our method performs well when compared to field-mapped floodplain initiation points, published flood risk maps, and digitised terrace surfaces. Our method works particularly well in higher relief areas, such as the Russian and South Fork Eel Rivers (CA), where the floodplain and terrace features are constrained within valleys. It is relatively insensitive to grid resolution, allowing the successful extraction of floodplain and terrace features at resolutions of 1 - 10 m.

Our new method has numerous applications in both the hydrological and geomorphological communities. It can allow the rapid extraction of floodplain features in areas where the data required for detailed hydrological modelling studies are unavailable, facilitating investigation of flood response, sediment transport, and alluviation. Furthermore, the automated extraction of terrace locations, heights, and other metrics could be used to examine the response of fluvial systems to climatic and tectonic perturbations, as well as the relative importance of lateral and vertical channel incision.

## 7 Software availability

Our software is freely available for download on GitHub as part of the Edinburgh Land Surface Dynamics Topographic Tools package at https://github.com/LSDtopotools. Full documentation on download, installation, and using the software can be found at http://lsdtopotools.github.io/LSDTT_book/

*Author contributions.* FJC, SMM, DTM, and DAV wrote the software for the feature extraction. MDH, LJS, and FJC collected the field data for floodplain validation; ABL collected the field data for terrace validation. FJC performed the analyses, created the figures, and wrote the manuscript with contributions from the other authors.

*Acknowledgements.* FJC is funded by the Carnegie Trust for the Universities of Scotland and NERC grant
NE/P012922/1. SMM is funded by NERC grant NE/P015905/1 and U.S. Army Research Office contract number W911NF-13-1-0478. DTM is funded by NERC grant NE/K01627X/1 and DAV is funded by NERC grant NE/L501591/1. LJS was supported by a NERC PhD studentship. ABL acknowledges support from the National Center for Earth-Surface Dynamics 2 Synthesis Postdoctoral Program. We are also grateful for additional finan-
cial support from the British Society for Geomorphology and the Royal Geographical Society with IBG. We would like to thank Stuart Grieve and Elizabeth Dingle for their help with fieldwork.

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

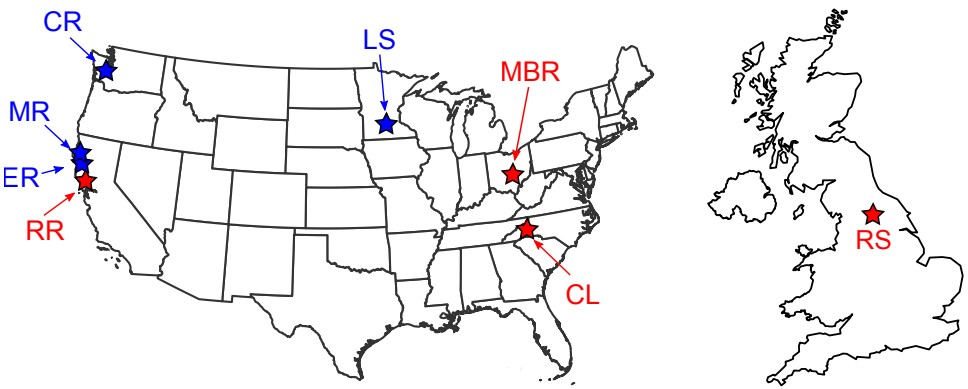

**Figure 1.** Maps of the US and UK showing the location of the eight field sites in the study. Red stars represent floodplain sites; blue stars represent terrace sites. RR = Russian River, CA; ER = South Fork Eel River, CA; MR = Mattole River, CA; CR = Clearwater River, WA; LS = Le Sueur River, MN; MBR = Mid Bailey Run, OH; CL = Coweeta Hydrologic Laboratory, NC; RS = River Swale, Yorkshire, UK.

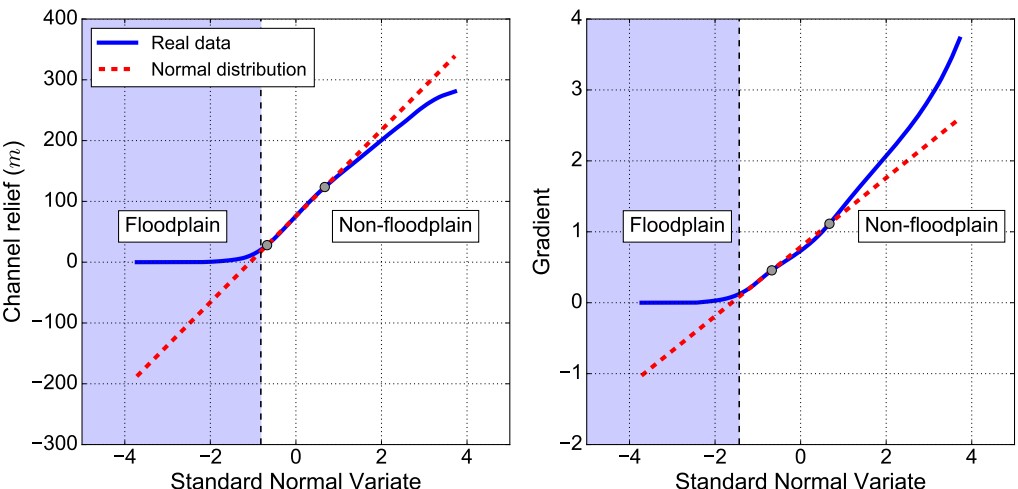

**Figure 2.** Example quantile-quantile plots for Mid Bailey Run, Ohio, showing probability density function of relief relative to the channel and slope. The probability density function of each is shown in blue, with the reference normal distribution shown by the red dashed line. The threshold (black dashed line) is selected where there is less than 1% difference between the real and reference distributions. The blue box highlights the portion of the distribution identified as floodplain.

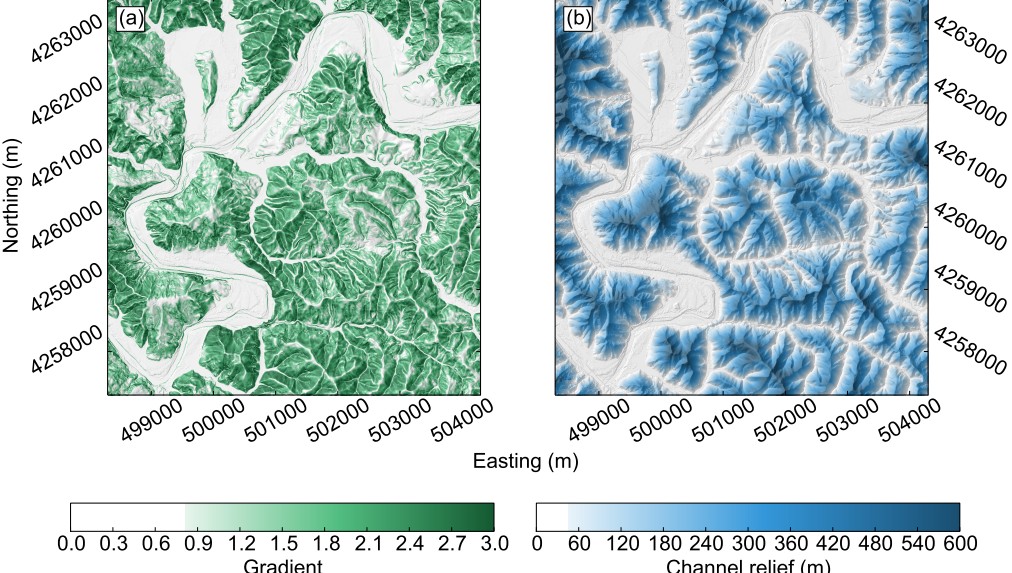

**Figure 3.** Maps showing a) gradient and b) relief relative to the nearest channel, $R_c$, for the Russian River field site. The areas of the landscape identified as below the threshold are shown in white, with values above the threshold then grading to darker colours. In order to be selected as floodplain, each pixel must be below the threshold for both gradient and $R_c$. The coordinate system is UTM Zone 10N.

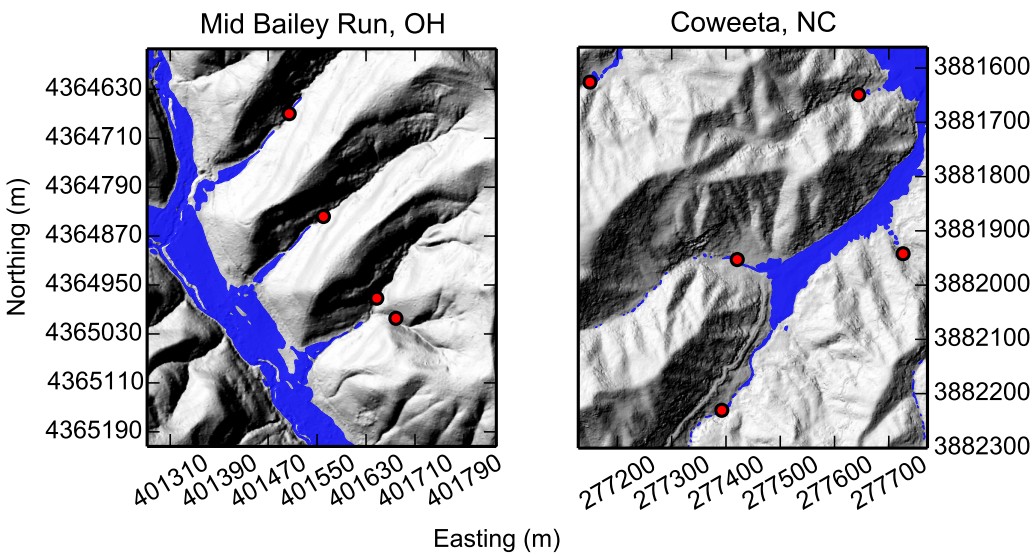

**Figure 4.** Shaded relief maps of Mid Bailey Run and Coweeta field sites showing the relationship between the predicted floodplain (blue) and the mapped floodplain initiation points (red). The UTM zone is 17N.

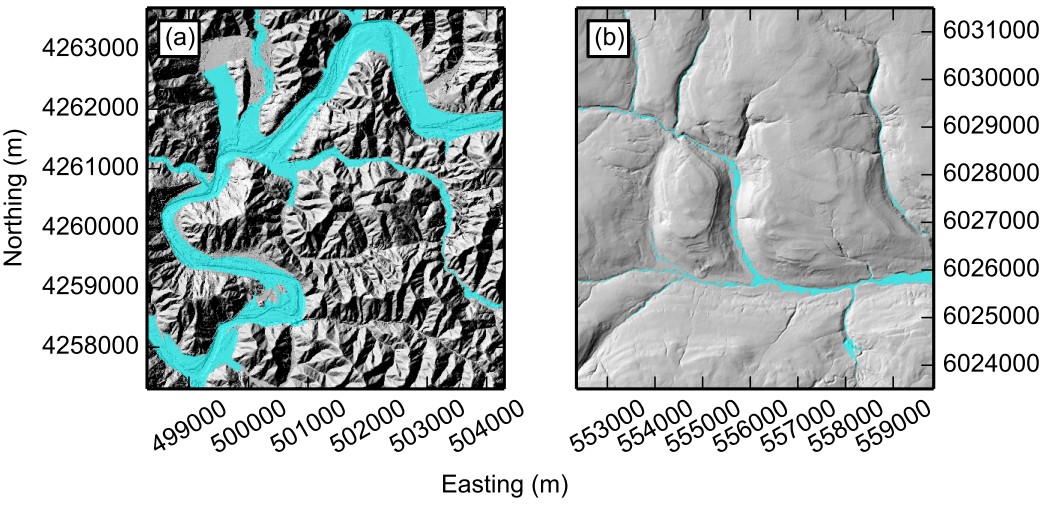

**Figure 5.** Shaded relief maps showing a) FEMA flood risk map for the Russian River, CA, UTM Zone 10N and b) EA flood risk map for the River Swale, UK, UTM Zone 30N. In some parts of the landscape the published flood maps do not extend all the way up the catchments.

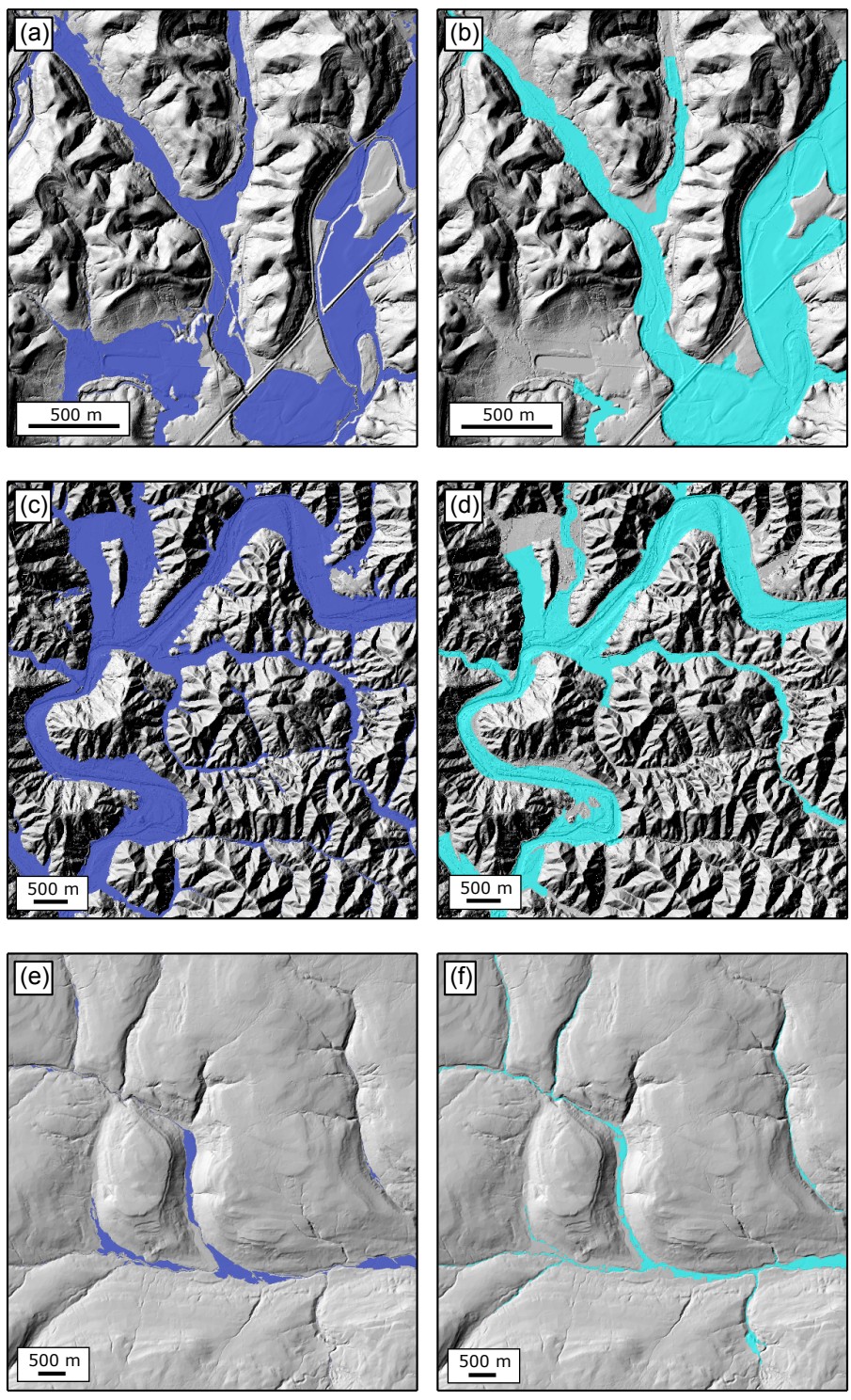

**Figure 6.** Shaded relief maps for each field site showing a comparison between the predicted floodplains (first column) and the published FEMA/EA maps (second column). (a) - (b) Mid Bailey Run, OH. (c) - (d) Russian River, CA. (e) - (f) River Swale, UK.

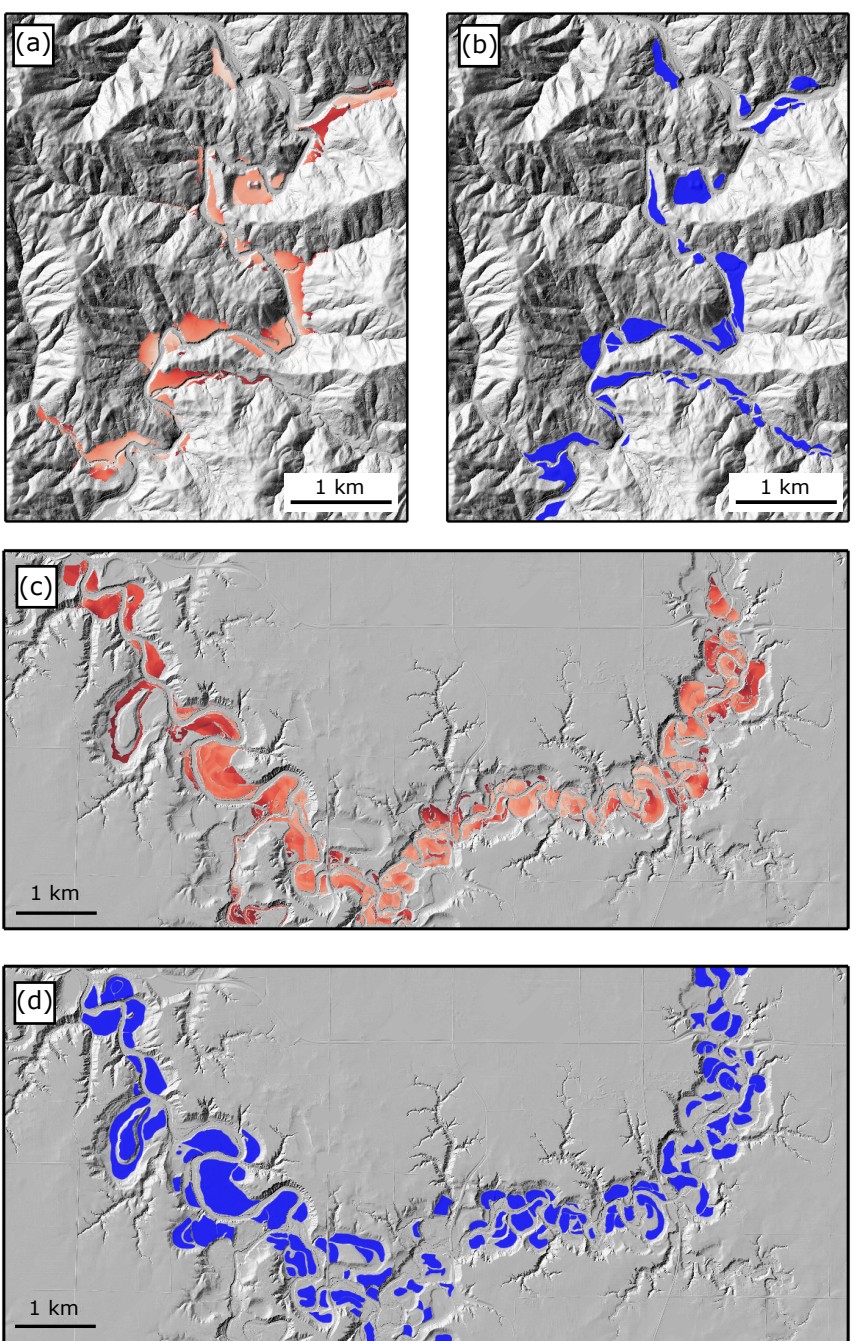

**Figure 7.** Shaded relief maps for the two field sites with LiDAR-derived DEMs showing a comparison between the predicted terraces (red) and the digitised terraces (blue). The predicted terraces are coloured by elevation compared to the channel, where darker red indicates higher elevation. (a) - (b) South Fork Eel River, CA. Maximum terrace height is 43 m. (c) - (d) Le Sueur River, MN. Maximum terrace height is 9.5 m.

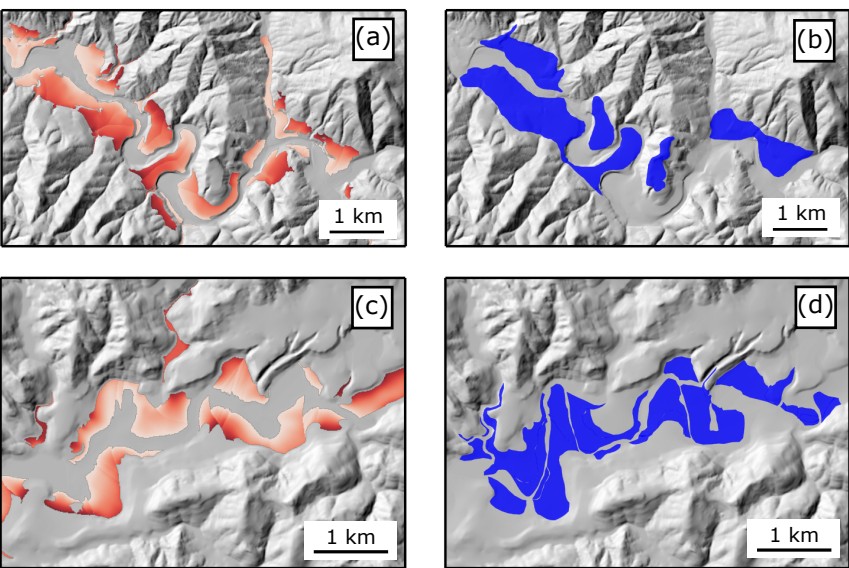

**Figure 8.** Shaded relief maps for the two field sites with 10 m resolution DEMs from the USGS NED showing a comparison between the predicted terraces (red) and the digitised terraces (blue). The predicted terraces are coloured by elevation compared to the channel, where darker red indicates higher elevation. (a) - (b) Mattole River, CA. Maximum terrace height is 50 m. (c) - (d) Clearwater River, WA. Maximum terrace height is 13 m.

**Table 1.** Channel relief and slope threshold for each field site

| Field site | Channel relief threshold | Slope threshold |
|---|---|---|
| Mid Bailey Run, OH | 23.69 | 0.15 |
| Coweeta, NC | 32.80 | 0.11 |
| Russian River, CA | 43.51 | 0.81 |
| River Swale, UK | 39.40 | 0.05 |
| South Fork Eel River, CA | 42.96 | 0.05 |
| Le Sueur River, MN | 9.42 | 0.05 |
| Mattole River, CA | 50.25 | 0.17 |
| Clearwater River, WA | 12.67 | 0.06 |

**Table 2.** Details of climate and lithology for each field site

| Field site | UTM Zone | MAP (mm) | MAT(°C) | Lithology | Comparison datasets | Grid res. (m) |
|---|---|---|---|---|---|---|
| Russian River, CA | 10°N | 1396 | 14.1 | Sandstones and shales, Quaternary alluvial deposits | FEMA flood hazard maps | 1 |
| Mid Bailey Run, OH | 17°N | 1005 | 10.9 | Sandstones, siltstones, shales | FEMA flood hazard maps Field-mapped FIPs | 1 |
| Coweeta, NC | 17°N | 1792 | 12.3 | Meta-sedimentary units | FEMA flood hazard maps Field-mapped FIPs | 1 |
| River Swale, UK | 30 °N | 898 | 8.4 | Limestones and sandstones | EA flood hazard maps | 5 |
| South Fork Eel River, CA | 10°N | 2009 | 12.7 | Greywackes and shales | Digitised terraces (Seidl and Dietrich, 1992) | 1 |
| Le Sueur River, MN | 15°N | 793 | 7.5 | Pleistocene tills and Ordovician dolostones | Digitised terraces (Gran et al., 2009) | 1 |
| Mattole River, CA | 10°N | 2593 | 12.8 | Sandstones and shales, Quaternary alluvial deposits | Digitised terraces (Dibblee and Minch, 2008; Limaye and Lamb, 2016) | 10 |
| Clearwater River, WA | 10°N | 3126 | 9.9 | Sandstones with interbedded shales | Digitised terraces (Wegmann and Pazzaglia, 2002; Limaye and Lamb, 2016) | 10 |

**Table 3.** Flow distances between the field-mapped FIPs and predicted floodplain extents

| Field site | Mapped FIP | Easting (m) | Northing (m) | Flow distance[1] |
|---|---|---|---|---|
| Mid Bailey Run, OH | T2FPI1 | 401513 | 4364940 | 59 |
| | T3FPI1 | 401622 | 4364773 | 85 |
| | T3FPI2 | 401661 | 4364732 | -49 |
| | WBT1FPI | 400090 | 4363977 | -23 |
| | WBT2FPI1 | 399865 | 4364215 | -1 |
| | T4FPI | 401342 | 4365472 | 28 |
| | T5FPI2 | 401072 | 4365675 | 0 |
| | T7FPI2 | 400670 | 4366152 | 2 |
| | T5FPI1 | 401208 | 4365807 | 0 |
| | T1FPI1 | 401443 | 4365150 | 0 |
| | TX3D3-FPI0 | 400718 | 4366277 | -42 |
| | TX3FPI1 | 400644 | 4366126 | -5 |
| | MBFPI | 400449 | 4366130 | -34 |
| | T7FPI1 | 400600 | 4366074 | -19 |
| | T4FPI2 | 401391 | 4365514 | 92 |
| | T6FPI1 | 400900 | 4365921 | -20 |
| Coweeta, NC | SF5 | 277212.380 | 3882554.000 | -51 |
| | BC1 | 276326.800 | 3880661.200 | -3 |
| | HCW | 277641.5 | 3881694.2 | 2 |
| | BC3 | 277584.633 | 3881138.653 | -3 |
| | HW1 | 278252.652 | 3881715.719 | 13 |
| | CB1 | 278089.041 | 3882301.638 | 12 |
| | HB1 | 277444.900 | 3882919.685 | -16 |
| | CC2 | 277098.745 | 3882348.108 | -2 |

[1] The distance between the mapped FIP and the upstream extent of the nearest floodplain patch predicted by our geomorphic method

**Table 4.** Results of the reliability ($r$), sensitivity ($s$), and overall quality ($Q$) analysis for each site

| Field site | Grid resolution (m) | $r$ | $s$ | $Q$ |
|---|---|---|---|---|
| Mid Bailey Run, OH | 1 | 0.73 | 0.76 | 0.59 |
| | 10 | 0.77 | 0.80 | 0.65 |
| Russian River, CA | 1 | 0.74 | 0.97 | 0.67 |
| | 10 | 0.70 | 0.96 | 0.68 |
| River Swale, UK | 5 | 0.84 | 0.65 | 0.58 |
| South Fork Eel River, CA | 1 | 0.65 | 0.72 | 0.52 |
| Le Sueur River, MN | 1 | 0.58 | 0.54 | 0.39 |
| Mattole River, CA | 10 | 0.58 | 0.65 | 0.44 |
| Clearwater River, WA | 10 | 0.56 | 0.55 | 0.39 |