# Peer review of "Geomorphometric delineation of floodplains and terraces from objectively defined topographic thresholds"

_Earth Surface Dynamics, 2017_

## Referee Comment (RC1) · Anonymous Referee #1 · 28 Apr 2017

The paper by Clubb et al. is an interesting and valid contribution to the journal. The authors propose a digital approach to mapping floodplains and terraces in different landscapes and compare their results with field measurements or flood maps derived from other sources. The paper is very well written and I enjoy reading it. Overall I think the authors provide a clear and detailed example of the validity of their procedure. However, I have few comments that I think might help to improve the paper.

1. First of all, I do appreciate the effort of creating an entirely automated procedure: this is the ultimate goal of many research, providing tools to avoid time consuming field surveys over large areas, in addition to allow understanding earth surface processes at the landscape scale. The paper states that prior approaches required manual editing

by the users, and they suggest their work is a step forward from these issues. They underline this fact many times in the manuscript, describing how their method is 'fully automated'. However, I think the authors should note that indeed, the procedure is still not fully automated. At page 7 - line 203: there is a suggested threshold, but such threshold can be changed by the user after visually inspecting the landscape - line 207: the user must provide the latitude and longitude do focus on a specific channel of interest (of course, in the case the user wants to focus on a specific channel on the whole landscape, which is understandable) - line 212: the user must specify the width of the swath, and this value can be estimated by a visual inspection of the DEMs. So it appears there is still some user-related parameters. I think, actually, what the authors propose is a procedure based on a fully automatic threshold (based on statistic) for the extraction (as the paper title correctly indicates). And statistic itself has been proven very useful in this task in many other research papers also in other fields, in addition to those mentioned by the authors in the introduction e.g. (Molly and Stepinski, 2007; Thommeret et al., 2010; Pelletier, 2013).

2. Table 3 reports the accuracy of the floodplain extraction. I tried to do the math myself but I do not get the value of 8m for the Mid Bailey Run. Maybe I am missing something? Also, the mean distance is not a reliable information, the authors errors arrive to values of ∼90 m. This measurement might not be that influent for landscape-scale processes, but for flood inundation maps, especially near human settlements, it might make a difference, so I think it is worth discussing it, unless the authors believe that this error is an outlier due to specific reasons (but it still might be worth mentioning it). Maybe they could evaluate reliability and sensitivity for the FIP (and not just for the overall floodplain extraction) as (Orlandini et al., 2011) did to assess the goodness of its point identification. This would also make the floodplain initiation point analysis consistent with the floodplain identification and terraces extraction analysis.

3. Lines from 285 to 335 should be in the method section. This is not a result, but rather the metrics the authors choose to evaluate the quality of their results. Concerning this

approach (also for the previous point), I think the use of an overall Quality measure would be appropriate, rather than just using reliability and sensitivity. Overall quality can be evaluated according to (Heipke et al., 1997), which is the first one proposing the sensitivity and reliability formulation. This would allow the authors also to compare their quality with other works about feature extraction in literature. I would also argue that reliability and sensitivity in their broad sense do not report an overall 'spatial correlation' between the datasets, as stated by the authors (line 365), but only a specific relation between either false negatives or true positives. Hence why I would suggest to use an overall measure as well.

4. Line 383: Floodplain inundation and alluviation changes through time. However I am not sure these changes would affect the geomorphological floodplain in the timeframe expressed by the authors (2-5 years' differences) unless significant events happened in that timeframe.

5. Results discussion. Can the author explain why their method performs better for floodplain delineations rather than for terraces? Is there a reason related to the method itself, or to the topography under analysis? is it related to the method they use to extract the channels? I think this is worth discussing more. Also, can the authors provide information about what influences the rate of TP or FN (so reliability and sensitivity, and eventually overall quality if they decide to evaluate it)? I think this is an important information to give, so users willingly to apply the proposed method in other areas can understand where to expect better or worse results.

6. Line 418- on. The authors state their method is relatively insensitive to grid resolution. However, their optimum value of reliability is obtained with a 5m DEM rather than for a 1 m DEM, and there are variations in reliability and sensitivity when changing the resolution: in some cases, the r and s are higher for the 10m DEM. I wonder if the authors have an idea on why this happens (maybe less noise on the 10m DEM that can influence their evaluations? Maybe too much noise on the 1m?). I think this part is also worth discussing a bit, since the procedure is available to the public, and users

might have different datasets (not necessarily Lidar at 1m). I understand the shifts in the two indices are low in magnitude, but I think discussing them makes sense.

7. Figures The figures are clear and well described. Just a curiosity: figure 8c and d: the predicted terrace is quite different from the digitised one in the central part of the river. From a visual inspection, this appears as a quite well define terrace, what is this difference's cause? Also, is it possible to have a map of an area showing both the identified terrace and floodplain?

  REFERENCES Heipke C, Mayer H, Wiedemann C, Jamet O. 1997. Automated reconstruction of topographic objects from aerial images using vectorized map information. International Archives of the Photogrammetry, Remote Sensing 23: 47–56 Molly I, Stepinski TF. 2007. Automatic mapping of valley networks on Mars. Computers & Geosciences 33: 728 Orlandini S, Tarolli P, Moretti G, Dalla Fontana G. 2011. On the prediction of channel heads in a complex alpine terrain using gridded elevation data. Water Resources Research 47 (2): W02538 DOI: 10.1029/2010WR009648 Pelletier JD. 2013. A robust, two-parameter method for the extraction of drainage networks from high-resolution digital elevation models (DEMs): Evaluation using synthetic and real-world DEMs. Water Resources Research 49 (1) DOI: 10.1029/2012WR012452 Thommeret N, Bailly JS, Puech C. 2010. Extraction of thalweg networks from DTMs: application to badlands. Hydrology and Earth System Sciences 14 (8): 1527–1536 DOI: 10.5194/hess-14-1527-2010

---

## Referee Comment (RC2) · A. Wickert (Referee) · 10 May 2017

This paper is a strong contribution to ESURF and a clear step in the right direction towards a mapping floodplains and terraces. I am particularly pleased about the idea of using the quantile–quantile plot approach, which provides a null hypothesis (in this case, normally-distributed topography) against which the landscape may be tested. The majority of my comments are in the paper itself, an annotated version of which is attached. However, I will include some more general points here:

1. First, I will echo the first reviewer in writing that there needs to be a more clearly-defined line between "fully automated" and "semi-automated". In other words:

[Figure]

define the realms within which your model is automated or is not. Currently, the lack of a well-defined separation undercuts the advances that you really have made by making it seem as if you overstate the work and making the focus on the "it isn't that far" rather than "it is a big step beyond prior work". I have read that Clubb et al. have responded to the first reviewer already in response to this general concern, so I will go on to a couple more specific points:

(a) One arbitrary piece is the decision about how wide of a swath should be used to search for terraces. To me, this highlights something that has long been on my "to do" list: a tool to automatically compute the widths of river valleys (see Shaw et al., 2008, for an analogous problem in coastlines). So I think that your use of an user-defined parameter is due to the lack of a tool that is outside your current scope, making this a placeholder for a better method!

(b) Your wording hints that there are problems in terrace identification when a river exists below a high plateau surface, and that these require some special parameter choices. This could also be aided by a tool to identify valley widths, but a more satisfying explanation about possible failure modes and ways around them – especially considering the range of upland topogrpahies from steep lands with ridges to flat upland plateaus – would be more satisfying.

2. Second, and related: I wonder why you chose a Gaussian distribution as the "landscape null hypothesis" from which you search for variations. I see the power in its simplicity, but do wonder whether you could replace the Gaussian distribution with the distribution expected from a stream-power-erosion plus hillslope-diffusion (I'll write it in a linear way here) simple model: $\partial z/\partial t = -k_{SP}A^m S^n + k_{HS}\nabla^2 z$. By integrating through time (e.g., numerically with a landscape evolution model), one can generate a non-Gaussian "landscape null hypothesis". This to me would seem a more powerful approach insofar as it represents what

is expected on the landscape in absence of floodplains and terraces, but does have sensitivity to the $k$ values chosen (or calibrated to the given landscape with another automated procedure). Nevertheless, I think that some of the by-hand "tweaking" with the quantile–quantile plots could be reduced by comparing the measured landscape against a more physically-based elevation distribution. To be clear: I am happy to see this paper published without changing its entire basis, but would feel remiss to not leave a record of this idea as a potential future avenue for improvement.

In both of these cases, I think that your approach is the right set of steps towards a process that is fully automated, and think that the places in which it is not fully automated serve to highlight areas in which advances are needed; such advances can lie outside of the scope of this paper.

[Figure]

**Supplement:**

[revised manuscript text omitted]

---

## Referee Comment (RC3) · Anonymous Referee #3 · 11 May 2017

This paper presents a new technique for mapping floodplains and terraces from digital elevation models. The paper is generally well written and the approach is both novel and useful. My biggest concern is the authors' claim that the tool is fully automated, when it does not really produce reliable maps in fully automated mode and would require users to manually edit maps to make them reliable, just as is the case with any of the other semi-automated techniques out there. I would suggest the authors tone down the somewhat disparaging comments regarding existing semi-automated techniques and at the same time tone down the sales pitch on their method being fully automated (just add a caveat that user interaction is needed to produce reliable maps). Aside from that concern and a few other minor question and suggestions below I believe the paper will make a nice contribution to ESD.

Lines 92-99: This explanation is not articulated well. I suggest revising, and perhaps condensing this section on Dodov and Foufoula-Georgiou. It seems to be a disproportionate amount of information compared to other studies discussed and the extent to which this information is utilized in the rest of the paper.

Line 113: Overprediction is a feature, not a bug. These are decidedly semi-automated approaches and it is a benefit if the automated portion of the tool slightly overpredicts because it is easy for the user to manually clip polygons.

Line 179: So in the end you use Optimal Weiner filter, correct? If so, why go into detail about Perona-Malik? I suggest either making a better connection between the two filters and explaining how the Perona-Malik equations relate to the Open Weiner filter, or reduce discussion on P-M and instead provide more detail on the OW filter.

Line 202: terrace should be terraces

Line 203: The authors don't provide any evidence that third order is a reasonable threshold. I have frequently seen terrace features on first and second order streams in places in the northeastern, Midwestern and western US. I suggest removing this arbitrary suggestion and simply explaining how the user should determine what the threshold should be for their particular landscape.

Lines 220-234: The authors spend a lot of time explaining quantile-quantile plots. Such explanations may be best left for textbooks as q-q plots are fairly routine, but I leave it to the authors to decide whether or not it is necessary to include. More importantly, I think it is important that the authors explain why it is reasonable to assume that local gradients would follow a Gaussian distribution and why deviations from Gaussian are likely to be transitions between process domains.

Line 240: In what way to do you mean 'connected to the modern channel'? Certainly terraces can abut the modern channel.

**ESurfD**
Line 296: How and why did you separate flood zones into 100 year and greater than 100 year flood risk? Just based on comparison with the FEMA maps? If so, are the FEMA maps necessarily reliable? Many would consider floodplains above the 100 year flood flood zone to be terraces. At what point do you make this distinction?

Table 4: The authors were somewhat disparaging about semi-automated approaches that have been developed earlier. Seeing these reliability and sensitivity values, I would suggest that the tool they have developed is no different. In comparisons with mapped terraces the tool is mapping a lot of false positives and false negatives. To map terraces reliably a user would need to manually edit these extensively...that's fine...it's to be expected, really...and that's why previous algorithms have claimed to be semi-automated. But I would urge the authors not to make claims about it being a fully automated process when the automated process fails to produce a reliable map.

Lines 445-450: I don't think the authors have made a strong case that their method produces reliable maps as a fully automated system. I agree that their method is a useful first cut, but this is no different from Stout and Belmont or any of the other semi-automated approaches mentioned in the paper.

Line 469: There are several other key papers that could be cited as examples of using terraces to quantify sediment budgets: Trimble, S. W. (1999). Decreased rates of alluvial sediment storage in the Coon Creek Basin, Wisconsin, 1975-93. Science, 285(5431), 1244-1246. Belmont, P., Gran, K. B., Schottler, S. P., Wilcock, P. R., Day, S. S., Jennings, C., ... & Parker, G. (2011). Large shift in source of fine sediment in the Upper Mississippi River. Environmental science & technology, 45(20), 8804-8810. Brown, A. G., Carey, C., Erkens, G., Fuchs, M., Hoffmann, T., Macaire, J. J., ... & Walling, D. E. (2009). From sedimentary records to sediment budgets: multiple approaches to catchment sediment flux. Geomorphology, 108(1), 35-47.

Line 474: Several key papers needed to substantiate this statement as well. Lots of examples, such as: Pazzaglia, F. J., & Brandon, M. T. (2001). A fluvial record of
long-term steady-state uplift and erosion across the Cascadia forearc high, western Washington State. American Journal of Science, 301(4-5), 385-431. Avouac, J. P., & Peltzer, G. (1993). Active tectonics in southern Xinjiang, China: Analysis of terrace riser and normal fault scarp degradation along the Hotan-Qira fault system. Journal of Geophysical Research: Solid Earth, 98(B12), 21773-21807. Viveen, W., Schoorl, J. M., Veldkamp, A., & Van Balen, R. T. (2014). Modelling the impact of regional uplift and local tectonics on fluvial terrace preservation. Geomorphology, 210, 119-135.

**ESurfD**

---

## Author Response (AR1)

THE UNIVERSITY *of* EDINBURGH
School of Geosciences

Fiona J. Clubb
*School of Geosciences*
*University of Edinburgh*
*Drummond Street*
*Edinburgh, EH8 9XP*
*Phone: +44 (0)131 650 9170*
*Email: f.clubb@ed.ac.uk*

Greg Hancock
Associate Editor, Earth Surface Dynamics

May 26, 2017

Dear Dr. Hancock,

Thank you for considering our manuscript 'Geomorphometric delineation of floodplains and terraces from objectively defined topographic thresholds'. We are grateful to the reviewers for providing constructive feedback and allowing us to improve the manuscript.

We have made significant changes to our manuscript following the comments we received. Throughout the manuscript we have clarified the sections of our method that are fully objective (the selection of the thresholds for channel relief and gradient), and those for which user-defined parameters are required. We have included an additional measure of quality ($Q$) into our analysis along with the reliability ($r$) and sensitivity ($s$) metrics, which allows the overall performance of the method to be compared more easily to other studies of automatic feature extraction in the literature. We have also expanded the paper to include more discussion of the results, including the impact of grid resolution and comparison between the performance of the method for floodplains and terraces. Alongside this, we have added in a new section to the discussion (Section 5.3) on future research needs and the development of fully-automated methods of geomorphic feature extraction.

Please find below detailed responses to the individual points raised by each of the reviewers, along with a version of our manuscript highlighting the changes we have made to answer the reviewer comments. Throughout our responses we refer to line numbers in our manuscript: these are the correct line numbers in the manuscript with the changes incorporated. We have endeavoured to address all concerns and return the manuscript in a publication-ready state.

Sincerely,

Fiona J. Clubb

**Reviewer 1**

*The paper by Clubb et al. is an interesting and valid contribution to the journal. The authors propose a digital approach to mapping floodplains and terraces in different landscapes and compare their results with field measurements or flood maps derived from other sources. The paper is very well written and I enjoy reading it. Overall I think the authors provide a clear and detailed example of the validity of their procedure. However, I have few comments that I think might help to improve the paper.*

We would like to thank the reviewer for their comments, and their positive response to our manuscript. We have edited our manuscript accordingly including expanding the discussion section, and including an overall quality measure to compare the results of our methods to the published datasets. Details of our responses to the individual comments are outlined below.

*1. First of all, I do appreciate the effort of creating an entirely automated procedure: this is the ultimate goal of many research, providing tools to avoid time consuming field surveys over large areas, in addition to allow understanding earth surface processes at the landscape scale. The paper states that prior approaches required manual editing by the users, and they suggest their work is a step forward from these issues. They underline this fact many times in the manuscript, describing how their method is fully automated. However, I think the authors should note that indeed, the procedure is still not fully automated. At page 7 - line 203: there is a suggested threshold, but such threshold can be changed by the user after visually inspecting the landscape - line 207: the user must provide the latitude and longitude do focus on a specific channel of interest (of course, in the case the user wants to focus on a specific channel on the whole landscape, which is understandable) - line 212: the user must specify the width of the swath, and this value can be estimated by a visual inspection of the DEMs. So it appears there is still some user-related parameters. I think, actually, what the authors propose is a procedure based on a fully automatic threshold (based on statistic) for the extraction (as the paper title correctly indicates). And statistic itself has been proven very useful in this task in many other research papers also in other fields, in addition to those mentioned by the authors in the introduction e.g. (Molly and Stepinski, 2007; Thommeret et al., 2010; Pelletier, 2013).*

We would like to thank the reviewer for their appreciation of our goal in the paper of creating a fully objective method of feature extraction. We agree that there are user-defined parameters which are set in the method, and we have stated this clearly when outlining our methodology (e.g. Lines 200 - 217). In order to make this clearer we have changed references to this throughout the text to highlight that the threshold selection is fully automated, but that the method does require some visual inspection of the DEM prior to running the analysis. We have also added in references to the studies suggested here by the reviewer:
Line 221-226: 'Many methods of channel extraction employ statistical selection of topographic thresholds (e.g. Lashermes et al., 2007; Thommeret et al., 2010; Passalacqua et al., 2010a; Pelletier, 2013; Clubb et al., 2014), but this has yet to be developed for the identification of floodplains or terraces. We identify thresholds for $R_c$ and $S$ using quantile-quantile plots, which have previously been used in the detection of hillslope-valley transitions (e.g. Lashermes et al., 2007; Passlacqua et al., 2010a).'

*2. Table 3 reports the accuracy of the floodplain extraction. I tried to do the math myself but I do not get the value of 8m for the Mid Bailey Run. Maybe I am missing something? Also, the mean distance is not a reliable information, the authors errors arrive to values of $\approx$90 m. This measurement might not be that influent for landscape scale processes, but for flood inundation maps, especially near human settlements, it might make a difference, so I think it is worth discussing it, unless the authors believe that this error is an outlier due to specific reasons (but it still might be worth mentioning it). Maybe they could evaluate reliability and sensitivity for the FIP (and not just for the overall floodplain extraction) as (Orlandini et al., 2011) did to assess the goodness of its point identification. This would also make the floodplain initiation point analysis consistent with the floodplain identification and terraces extraction analysis.*
In response to this comment we have calculated the reliability and sensitivity of the method compared to the mapped FIPs instead of reporting the mean distance, in order to make the comparison more robust and to keep it consistent with the analysis for the rest of the data. The reliability and sensitivity values for Coweeta and Mid Bailey Run are reported on Lines 316-320. We have also added a discussion of the reliability and sensitivity compared to the mapped FIPs on Lines 392-400.

*3. Lines from 285 to 335 should be in the method section. This is not a result, but rather the metrics the authors choose to evaluate the quality of their results. Concerning this approach (also for the previous point), I think the use of an overall Quality measure would be appropriate, rather than just using reliability and sensitivity. Overall quality can be evaluated according to (Heipke et al., 1997), which is the first one proposing the sensitivity and reliability formulation. This would allow the authors also to compare their quality with other works about feature extraction in literature. I would also argue that reliability and sensitivity in their broad sense do not report an overall spatial correlation between the datasets, as stated by the authors (line 365), but only a specific relation between either false negatives or true positives. Hence why I would suggest to use an overall measure as well.*

We have moved this section from the results to the methodology (Section 2.3, Comparison with published data). In terms of the quality analysis, we would argue that using the reliability and sensitivity values does allow comparison of the quality with other works in the literature: numerous studies presenting methods of feature extraction have reported the reliability and sensitivity, for example in channel extraction (e.g. Orlandini et al., 2011; Clubb et al., 2014) and in floodplain identification (e.g. Manfreda et al., 2014). In order to determine the performance of the method spatially, we also report the flow distance between the mapped and predicted floodplain initiation points for the field mapped data (Table 3). However, for the terraces and the published flood maps, metrics of length do not provide a good predictor of the performance of the method, therefore we decided to report reliability and sensitivity values which take into account the true or false positives or negatives based on the entire DEM. However, as suggested, we have also added in the overall quality analysis based on Heipke et al. (1997) for each of the comparisons (Table 4).

*4. Line 383: Floodplain inundation and alluviation changes through time. However I am not sure these changes would affect the geomorphological floodplain in the timeframe expressed by the authors (2-5 years differences) unless significant events happened in that timeframe.*

We have removed this sentence from the discussion - although it's interesting to note the timescales of the formation of geomorphic floodplains are not well understood. A potential application of our method could be to compare the different floodplains predicted geomorphically with those of a specific magnitude event predicted through hydrological modelling.

*5. Results discussion. Can the author explain why their method performs better for floodplain delineations rather than for terraces? Is there a reason related to the method itself, or to the topography under analysis?is it related to the method they use to extract the channels? I think this is worth discussing more. Also, can the authors provide information about what influences the rate of TP or FN (so reliability and sensitivity, and eventually overall quality if they decide to evaluate it)? I think this is an important information to give, so users willingly to apply the proposed method in other areas can understand where to expect better or worse results.*

The sites used for terrace identification were generally lower relief than those for the floodplain extraction, which is a potential reason for the worse performance of the method in these sites. In the terrace site with higher relief (South Fork Eel River), the method performed as well as for the floodplain identification. The method of channel extraction will not influence the results of the algorithm, as we only extract floodplains or terrace on higher order channels which are not affected by the locations of the first order channels. We have expanded the discussion to include a section on comparison of the performance of the method between floodplain and terrace extraction:

Lines 465-469: 'The results of the quality analysis for the eight field sites (Table 4) showed that the method performed better in the floodplain identification compared to the terrace identification. This may be due to the fact that, with the exception of the South Fork Eel River, the sites used for terrace extraction are lower relief than those used to test the floodplain extraction (e.g. Figures 6 - 8).'

Our manuscript includes information in the discussion about potential influences on the reliability, sensitivity, and overall quality for both the floodplain extraction (Lines 406-422) and terrace extraction

(throughout Section 5.2). We have also included discussion of the types of landscape in which the method may work best:

Lines 456-460: 'As our method relies on the distribution of relief relative to the channel in order to select the threshold for terrace identification, it will work best in areas where there is a greater contrast between the slope and relief of the terrace surfaces compared to the surrounding topography. This is similar to other semi-automated terrace extraction methods (e.g. Stout and Belmont, 2014; Hopkins and Snyder, 2016).'

*6. Line 418- on. The authors state their method is relatively insensitive to grid resolution. However, their optimum value of reliability is obtained with a 5m DEM rather than for a 1 m DEM, and there are variations in reliability and sensitivity when changing the resolution: in some cases, the r and s are higher for the 10m DEM. I wonder if the authors have an idea on why this happens (maybe less noise on the 10m DEM that can influence their evaluations? Maybe too much noise on the 1m?). I think this part is also worth discussing a bit, since the procedure is available to the public, and users might have different datasets (not necessarily Lidar at 1m). I understand the shifts in the two indices are low in magnitude, but I think discussing them makes sense.*

We have expanded the discussion to suggest potential reasons why grid resolution may cause some small changes in the values of reliability, sensitivity, and overall quality:

Lines 441-445: 'High-resolution topographic data may contain both small-wavelength topographic noise caused by tree throw and biotic activity (Roering et al., 2010; Marshall and Roering, 2014), as well as synthetic noise from point cloud processing (Liu, 2008; Meng et al., 2010). This noise may affect the calculation of topographic metrics (Grieve et al., 2016c), potentially leading to differences in the location of extracted floodplains or terraces compared to the lower resolution data.'

*7. Figures The figures are clear and well described. Just a curiosity: figure 8c and d: the predicted terrace is quite different from the digitised one in the central part of the river. From a visual inspection, this appears as a quite well define terrace, what is this differences cause? Also, is it possible to have a map of an area showing both the identified terrace and floodplain?*

We think that the very subtle change in elevation between the different terraces in the central part of the valley in this landscape compared to the others (e.g. the Mattole River, Figs 8a and b) makes it difficult to identify these accurately compared to the digitised terraces. If the terraces are very close in elevation to the modern floodplain then it can be difficult to distinguish between these from the DEM alone: we think that some of the portions of the landscape identified as digitised terraces may be selected as modern floodplain in our method. We tested the ability of the method to distinguish between floodplains and terraces, and found that the best way of separating between floodplains and terraces is to use a threshold height of the terraces above the modern river channel, which is user-defined, as in lower resolution DEMs patches of predicted floodplain/terrace may be connected. We have added in a section to the discussion of the problems of distinguishing between modern floodplain and terraces:

Lines 470-478: 'Another potential cause of error between the predicted and digitised terrace locations may be problems in distinguishing whether features represent the modern floodplain or terraces. In our method a minimum height above the modern channel is set, where pixels above this height are classified as terrace, and below this height are classified as floodplain. In some cases, particularly where the terraces are at a similar elevation to that of the modern channel, our method may mistakenly identify terraces as being part of the modern floodplain, or vice versa. An example of this may be the Clearwater River site, where our method had lower values of the quality metrics (Figures 8c and d, Table 4). In this site, the digitised terraces are close in elevation to the modern channel, with a maximum terrace height of 13 m.'

We have not included the combined floodplain and terrace maps into the paper, as we want to keep the sections on floodplain and terrace identification separate to fit in with the overall structure of the paper. However, we include the combined map for the Clearwater River here for reference.

[Figure]

*Shaded relief map of the Clearwater River, WA, showing combined floodplain (blue) and terrace (red) map. Where terraces are close to the elevation of the modern channel it can be difficult to distinguish between terraces and the active floodplain.*

**Reviewer 2: Andrew Wickert**

*This paper is a strong contribution to ESURF and a clear step in the right direction towards a mapping floodplains and terraces. I am particularly pleased about the idea of using the quantilequantile plot approach, which provides a null hypothesis (in this case, normally-distributed topography) against which the landscape may be tested. The majority of my comments are in the paper itself, an annotated version of which is attached.*

**General comments**

*1. First, I will echo the first reviewer in writing that there needs to be a more clearly-defined line between "fully automated" and "semi-automated". In other words: define the realms within which your model is automated or is not. Currently, the lack of a well-defined separation undercuts the advances that you really have made by making it seem as if you overstate the work and making the focus on the "it isnt that far" rather than "it is a big step beyond prior work". I have read that Clubb et al. have responded to the first reviewer already in response to this general concern, so I will go on to a couple more specific points:*

In response to this comment, and the comments from Reviewer 1, we have edited our manuscript to highlight the distinction between the parts of our method that are fully automated (the statistical selection of the thresholds from the quantile-quantile plots) and the need for some user-defined parameters. Our method does still have some parameters that are user-defined (threshold stream order for running on a landscape scale, width of the swath, and minimum height above the channel for floodplain/terrace distinction). However, in general these parameters can be estimated easily by the user from visual inspection of the DEM, and don't require the input of any independent datasets, unlike previous methods. However we agree that future research is needed in order to create a fully autonomous method, which is beyond the scope of our paper at the moment. We have added a section

to the discussion on future research directions, highlighting the points raised in the review comments: Section 5.3: 'A key goal for the Earth surface research community is to develop fully-automated methods of feature extraction from DEMs in order to avoid expensive and time consuming field-mapping, and to investigate the controls on geomorphic processes at a landscape scale. Our new method of floodplain and terrace delineation attempts to meet some of these research needs, by allowing the statistical determination of the thresholds for feature extraction. However, our method still requires the input of some user-defined parameters. If the method is run across the whole landscape, the user must set a threshold stream order for the calculation of elevation compared to the nearest channel. This is necessary so that each pixel is mapped to the main channel along which floodplains or terraces have formed, rather than narrow tributary valleys. This threshold can be determined by the user based on a visual inspection of the DEM compared to the channel network. If the user runs the method based on the swath mode, the width of the swath profile must be set. This can also be done based on a visual inspection of the DEM to provide a sufficiently wide swath compared to the valleys in the landscape. Furthermore, if the method is run in the swath mode, then a minimum terrace height must be set in order to delineate between floodplains and fluvial terraces.

However, future development of new algorithms, such as extraction of valley widths, would allow these parameters to be set based on the topographic data alone. Our method represents a first step towards this goal of fully-automated geomorphic feature identification, which can be improved upon with future research. The combination of different algorithms for terrain analysis, such as hillslope flow routing, channel network extraction, floodplains, and fluvial terraces, would allow an objective landscape-scale investigation of the controls on geomorphic processes.'

*(a) One arbitrary piece is the decision about how wide of a swath should be used to search for terraces. To me, this highlights something that has long been on my "to do" list: a tool to automatically compute the widths of river valleys (see Shaw et al., 2008, for an analogous problem in coastlines). So I think that your use of an user-defined parameter is due to the lack of a tool that is outside your current scope, making this a placeholder for a better method!*
This is definitely an area that needs further research, and would improve our method along with other algorithms for digital terrain analysis. We have added this to our new section in the discussion (see reply to general comment above).

*(b) Your wording hints that there are problems in terrace identification when a river exists below a high plateau surface, and that these require some special parameter choices. This could also be aided by a tool to identify valley widths, but a more satisfying explanation about possible failure modes and ways around them  especially considering the range of upland topographies from steep lands with ridges to flat upland plateaus  would be more satisfying.*
Yes, the method does not work in areas as well where there is less distinction between the relief structure of the surrounding topography (for example, the plateau surface in the Le Sueur River site) compared to the floodplains or terraces. We have added in some more discussion of the results for the Le Sueur River to clarify the difficulties of automatic feature extraction in these landscapes:
Lines 460-465: 'The Le Sueur River is currently incising through Pleistocene tills, forming a low-gradient surface or plateau (Fisher, 2003; Gran et al., 2009; Belmont et al., 2011). High-altitude, low-gradient surfaces, such as relict plateaus, may result in error in the method due to the difficulty in distinguishing the distribution of terrace elevations from these low-relief surfaces. The Le Sueur River basin is also heavily influenced by human land use, which makes feature extraction challenging (Passalacqua et al., 2012).'

*2. Second, and related: I wonder why you chose a Gaussian distribution as the 'landscape null*

*hypothesis' from which you search for variations. I see the power in its simplicity, but do wonder whether you could replace the Gaussian distribution with the distribution expected from a stream-power-erosion plus hillslope-diffusion (I'll write it in a linear way here) simple model: $\delta z/\delta t = -k_{SP}A^m S^n + k_{HS} \bigtriangledown^2 z$. By integrating through time (e.g., numerically with a landscape evolution model), one can generate a non-Gaussian 'landscape null hypothesis'. This to me would seem a more powerful approach insofar as it represents what is expected on the landscape in absence of floodplains and terraces, but does have sensitivity to the k values chosen (or calibrated to the given landscape with another automated procedure). Nevertheless, I think that some of the by-hand "tweaking" with the quantile-quantile plots could be reduced by comparing the measured landscape against a more physically-based elevation distribution. To be clear: I am happy to see this paper published without changing its entire basis, but would feel remiss to not leave a record of this idea as a potential future avenue for improvement.*

We chose the Gaussian distribution to use as the reference distribution for the elevation distribution in the landscape in order to keep the approach general, so that it could be applied across multiple landscapes with varying relief, and to limit the amount of user-defined parameters in the method as much as possible. Furthermore, the Gaussian distribution has also been used in feature extraction algorithms previously (e.g. Lashermes et al., 2007; Passalacqua et al., 2010). The idea of using a simple stream power and hillslope diffusion model to generate a distribution of elevation and slopes as a 'null hypothesis' is a very interesting one, which we could potentially apply to improve our method in the future. However, it may actually generate more user-defined parameters than the Gaussian distribution does at the moment (as you say, this may be sensitive to both erodibility and hillslope diffusivity). Although we feel it is beyond the scope of the paper to add this in at the moment, we have expanded our discussion to include some more of the potential limitations of using the Gaussian distribution to model relief.

Lines 480-489: 'This may be the case if the threshold for elevation compared to the channel selected by the quantile-quantile plot is lower than that of the highest terrace elevations. This can be examined for the landscape in question by a visual inspection of the quantile-quantile plots and the location of the threshold compared to the distribution of channel relief (e.g. Figure 2). Our method fits a Gaussian distribution to the quantile-quantile plots, and selects the thresholds as the deviation of the real data from this distribution, as a simple general model of elevation distributions that can be applied across multiple landscapes. However, in some landscapes, the distribution of elevations may not be accurately represented by a Gaussian distribution. A future avenue for development of this method may be to include multiple models for elevation distributions from which to select the thresholds of elevation and gradient.'

*In both of these cases, I think that your approach is the right set of steps towards a process that is fully automated, and think that the places in which it is not fully automated serve to highlight areas in which advances are needed; such advances can lie outside of the scope of this paper.*

This is a good point - again we have tried to address this by adding in our new section to the discussion (Section 5.3).

**Specific comments**

*Line 130: I'm interested to see how well the method of selecting breaks in the quantile/quantile plots performs, especially in landscapes with differing hypsometries*
See response to general comment 2.

*Line 164: I understand what you mean, but not how you wrote it*
We have reworded this sentence to make it clearer:
Line 162: 'Following the methodology of Passalacqua et al. (2010a), we set the number of iterations ($t$)

to 50 and the calculation of $\lambda$ as the 90% quantile.'

*Line 205: Does this take away from the full automation? Maybe "much more fully automated" but not completely? (I can also imagine that you see this as negligible compared to other factors that led to more emphasis on the "semi" part of "semi-automation".)*
See response to general comment 1.

*Line 214: This also leads to more 'semi' automation, but I don't think this is a bad thing. The problem of automating a method to find the width of a valley is, in itself, an unsolved problem!*
Again, we have added in a section to the discussion about the user-defined parameters, and suggested that a future research need is a method to automatically determine valley width, although in order to get a representative hypsometry distribution for the quantile-quantile plots, the width of the swath needs to include the hillslopes on either side of the valley. It should represent the full length scale of the ridge-to-valley topography. Again, this is a difficult problem to determine automatically from the topography alone.

*Line 231: Is there any landscape characteristic that you could use to help guide which values of these to pick? Otherwise I worry that you are undercutting your point that you have a fully-automated method.*
At the moment, the user can set the values of these percentiles for the reference Gaussian distribution. The best way to do this is to visually assess the quantile-quantile plots for the landscape (e.g. Figure 2), and determine whether the Gaussian distribution is fit through the real data appropriately. We have added this in to the discussion (see answer to general comment 2).

*Line 250: You jump into 'two of the four', etc. below; could you tell the reader which are included in each cluster of four? (Addendum: you define the second cluster of four for terraces, below, but not the first cluster of four for floodplain extent.)*
Done

*Line 428: This is probably just a sign of the times changing, but when I hear "USGS NED", I think of the digitized contours and the associated stair-step errors. Could you clarify if these are from SRTM, perhaps above at the first introudction of the NED?*
The DEMs used in the study were from the USGS NED as stated our manuscript: the NED comes from a variety of sources, including SRTM, and LiDAR, which is merged together into a seamless DEM for the US. It is now updated on a two-monthly cycle to include new elevation data as it is produced. In some places we think it may still use digitised contours, which does have its problems, but it is difficult to determine where this is the case. We have added in a sentence clarifying the source datasets for the USGS NED:
Line 355: 'The USGS NED is a seamless dataset created for the conterminous US, using a variety of elevation products which is updated on a two-month cycle.'

*Line 436: Is the issue that the Des Moines Lobe till surface (flat!) that lies above the incised valley confounds attempts to look just at the valley bottom? In other words – will the algorithm in general perform poorly on incised-plateau landscapes, and well on landscapes with ridges and no high flat surfaces that are not terraces?*
We have added in some extra discussion of the limitations of the terrace method in the Le Sueur site, and generally in areas where there may be relict plateaus or low gradient surfaces that are not terraces (see answer to general comment 1(b)).

*Line 445: As I was thinking – fully-automated within a specific scope.*
See answer to general comment 1.

*Line 479: I think you still need to define better the scopes within which it is objective, or how it is better than the previous methods. I like the quantile-quantile approach, but do have concerns that the user's ability to pick better intercepts 'by eye' begs a clear definition between the objective and subjective portions of your metrics. It also opens the door to questions regarding the subjectivity, which may result from other landscape metrics not yet being developed.*
We have added in extra clarification of the automation of the method and the setting of the user defined parameters (answer to general comment 1, and Section 5.3 of the revised manuscript). We have also added in extra discussion of the quantile-quantile plots and the potential limitations of using the Gaussian distribution to fit as a reference, in answer to general comment 2.

*Line 484: rm preceding comma if this clause is related only to "swath profile"*
Done

*Table 2: dolostones [this is picky, but "dolomite" is the mineral even though it is colloquially used for the rock]*
Done

*Table 2: Was this a placeholder?*
Yes...we have changed this in the revised manuscript!

**Reviewer 3**

*This paper presents a new technique for mapping floodplains and terraces from digital elevation models. The paper is generally well written and the approach is both novel and useful. My biggest concern is the authors claim that the tool is fully automated, when it does not really produce reliable maps in fully automated mode and would require users to manually edit maps to make them reliable, just as is the case with any of the other semi-automated techniques out there. I would suggest the authors tone down the somewhat disparaging comments regarding existing semi-automated techniques and at the same time tone down the sales pitch on their method being fully automated (just add a caveat that user interaction is needed to produce reliable maps). Aside from that concern and a few other minor question and suggestions below I believe the paper will make a nice contribution to ESD.*
Thank you very much for your comments on our manuscript. In response to your concerns, along with those from the other two reviewers, we have made a clear distinction in the paper about which part of the method are automated, and which parts still require user-defined parameters. We did not intend to be dismissive of other techniques of identifying floodplains and terraces - we agree that these methods are very useful, and have stated this in our manuscript. We have tried to build on these methods by developing statistical techniques for the selection of the thresholds in our method of elevation compared to the channel and local gradient. We have made clearer in our discussion that we believe that the different methods are valuable depending on the scale of the analysis, as well as the field site from which the floodplains/terraces are being extracted.

*Lines 92-99: This explanation is not articulated well. I suggest revising, and perhaps condensing this section on Dodov and Foufoula-Georgiou. It seems to be a disproportionate amount of information compared to other studies discussed and the extent to which this information is utilized in the rest of the paper.*
We have condensed this section as suggested.

*Line 113: Overprediction is a feature, not a bug. These are decidedly semi-automated approaches and it is a benefit if the automated portion of the tool slightly overpredicts because it is easy for the user to manually clip polygons.*

We have added in a sentence here to state that the user can manually clip the over-predicted surfaces and remove areas selected incorrectly:

Line 108-111: 'These semi-automated methods allow the user to manually clip over-predicted terrace surfaces based on field data and DEM observations, and remove selected surfaces that do not represent terraces, such as roads, alluvial fans, or water bodies (Stout and Belmont, 2014)'

*Line 179: So in the end you use Optimal Weiner filter, correct? If so, why go into detail about Perona-Malik? I suggest either making a better connection between the two filters and explaining how the Perona-Malik equations relate to the Open Weiner filter, or reduce discussion on P-M and instead provide more detail on the OW filter.*

We use the Perona-Malik filter for the method of floodplain/terrace extraction. The Perona-Malik filter is a non-linear filter which enhances the transition between features, such as hillslopes/valleys, while preferentially smoothing low gradient surfaces, such as floodplains or terraces. The Optimal Wiener filter is only used here for the extraction of the channel networks using the method outlined by Grieve et al. (2016, ESURF). We have added a sentence to clarify this in the manuscript:

Line 173: 'The Optimal Wiener filter is only used to extract the channel network: we use the Perona-Malik filtering to extract the floodplains and terraces.'

*Line 202: terrace should be terraces*
Done

*Line 203: The authors dont provide any evidence that third order is a reasonable threshold. I have frequently seen terrace features on first and second order streams in places in the northeastern, Midwestern and western US. I suggest removing this arbitrary suggestion and simply explaining how the user should determine what the threshold should be for their particular landscape.*

In each of our field sites we found that a third order threshold was appropriate for where the terraces initiated in the landscape (see Figures 7 and 8). We have changed this section to state this, and we have clearly stated that a visual inspection of the DEM compared to the channel network should allow the user to select the appropriate threshold stream order:

Lines 203-207: 'We found that a threshold of third order channels was appropriate for each of our field sites, based on a visual inspection of the DEM. One of the outputs of our software package is a raster of the channel network labelled by the Strahler stream order. The user can identify an appropriate threshold stream order based on visual inspection of floodplain and terrace surfaces compared to this network.'

*Lines 220-234: The authors spend a lot of time explaining quantile-quantile plots. Such explanations may be best left for textbooks as q-q plots are fairly routine, but I leave it to the authors to decide whether or not it is necessary to include. More importantly, I think it is important that the authors explain why it is reasonable to assume that local gradients would follow a Gaussian distribution and why deviations from Gaussian are likely to be transitions between process domains.*

We believe that it is important to include the description of the quantile-quantile plots as this is a key part of our methodology for selecting the thresholds of gradient and elevation compared to the channel from the DEMs. We chose a Gaussian distribution as a simple model, which can be applied generally over a range of landscapes, and has been used in previous methods of feature extraction (Lashermes et al., 2007; Passalacqua et al., 2010). We have added in some more discussion about the Gaussian distributions in response to this comment plus comments from Reviewer 2.

Lines 478-489: 'Furthermore, in some cases our method did not select all of the terraces identified by the field mapping, particularly at the highest elevations compared to the modern channel (e.g. Figure 7c and d). This may be the case if the threshold for elevation compared to the channel selected by the quantile-quantile plot is lower than that of the highest terrace elevations. This can be examined for the landscape in question by a visual inspection of the quantile-quantile plots and the location of the threshold compared to the distribution of channel relief (e.g. Figure 2). Our method fits a Gaussian distribution to the quantile-quantile plots, and selects the thresholds as the deviation of the real data from this distribution, as a simple general model of elevation distributions that can be applied across multiple landscapes. However, in some landscapes, the distribution of elevations may not be accurately represented by a Gaussian distribution. A future avenue for development of this method may be to include multiple models for elevation distributions from which to select the thresholds of elevation and gradient.'

*Line 240: In what way to do you mean connected to the modern channel? Certainly terraces can abut the modern channel.*
The method identifies patches of floodplain as those which are at a similar elevation to the modern channel (based on the extracted channel network), whereas terraces should be at a higher elevation compared to the channel. This was not clear in our original wording: we have rephrased this and added more discussion about the separation between floodplains and terraces to the manuscript based on comments from Reviewer 1.

*Line 296: How and why did you separate flood zones into 100 year and greater than 100 year flood risk? Just based on comparison with the FEMA maps? If so, are the FEMA maps necessarily reliable? Many would consider floodplains above the 100 year flood flood zone to be terraces. At what point do you make this distinction?*
The separation of flood zones into 100-year and greater than 100 year was on the FEMA maps which are classified based on the annual percentage chance of flooding. There may be some errors with the FEMA flood maps based on this: this may a cause of some of the discrepancies between the floodplains extracted from our method and with these published maps. We have a section in our manuscript discussing some of the potential problems with the FEMA flood maps:
Lines 423-431: 'Published flood maps are useful in providing an independent estimate of likely floodplains in each field site. However, there are potential limitations to these maps which must be carefully considered, and may result in some of the differences compared to geomorphic floodplain prediction techniques. Hydrodynamic models have a large number of parameters, which require careful calibration with field and hydraulic data, such as channel roughness and discharge data from gauging stations. Furthermore, due to the time-consuming and expensive nature of these studies, flood maps are often not produced for small catchment sizes, and may therefore be incomplete on a landscape-scale (e.g. Figure 4). There may also be differences in the methodology used in producing these maps for each site, depending on the input topographic data and modelling software used.'
The distinction between floodplains and terraces is something that may also cause some problems in our method, especially when the terraces are close in elevation to the modern channel. We have also added in more discussion about this to our manuscript based on comments from Reviewer 1.

*Table 4: The authors were somewhat disparaging about semi-automated approaches that have been developed earlier. Seeing these reliability and sensitivity values, I would suggest that the tool they have developed is no different. In comparisons with mapped terraces the tool is mapping a lot of false positives and false negatives. To map terraces reliably a user would need to manually edit these extensively...thats fine...its to be expected, really...and thats why previous algorithms have claimed to be semi-automated. But I would urge the authors not to make claims about it being a fully automated process when the*

*automated process fails to produce a reliable map.*

We did not intend at all to be disparaging about semi-automated approaches that have been previously developed: we think these methods are very useful, particularly in areas where there is some field data available to calibrate the selection of thresholds and user-defined parameters. We have tried to build on these methods by developing statistical techniques for the selection of the thresholds in our method of elevation compared to the channel and local gradient. As previously stated, we have now made a clear distinction in our manuscript between the user-defined parameters and these thresholds which are calculated statistically. We have made clearer in our discussion that we believe that the different methods are valuable depending on the scale of the analysis in question and location from which the floodplains/terraces are being extracted:

Lines 492-497: 'Semi-automated methods of terrace identification, where the terrace polygons are manually edited by the user, are particularly useful in areas where independent datasets of terrace locations are available for calibration, and may be more appropriate than our method on site-specific scales (e.g. Stout and Belmont, 2014). However, the selection of thresholds based on a objective statistical approach means that our method can be applied in areas where these data do not exist, on a broader landscape scale, or as a rapid first-order predictor of terrace locations.'

*Lines 445-450: I dont think the authors have made a strong case that their method produces reliable maps as a fully automated system. I agree that their method is a useful first cut, but this is no different from Stout and Belmont or any of the other semi-automated approaches mentioned in the paper.*

See reply to comment above.

*Line 469: There are several other key papers that could be cited as examples of using terraces to quantify sediment budgets: Trimble, S. W. (1999). Decreased rates of alluvial sediment storage in the Coon Creek Basin, Wisconsin, 1975-93. Science, 285(5431), 1244-1246. Belmont, P., Gran, K. B., Schottler, S. P., Wilcock, P. R., Day, S. S., Jennings, C., ... & Parker, G. (2011). Large shift in source of fine sediment in the Upper Mississippi River. Environmental science & technology, 45(20), 8804- 8810. Brown, A. G., Carey, C., Erkens, G., Fuchs, M., Hoffmann, T., Macaire, J. J., ... & Walling, D. E. (2009). From sedimentary records to sediment budgets: multiple approaches to catchment sediment flux. Geomorphology, 108(1), 35-47.*

We have added in the suggested references.

*Line 474: Several key papers needed to substantiate this statement as well. Lots of examples, such as: Pazzaglia, F. J., & 
[revised manuscript text omitted]